# A parsimonious neutral model suggests Neanderthal replacement was determined by migration and random species drift

Oren Kolodny [1] & Marcus W. Feldman[1]

Most hypotheses in the heated debate about the Neanderthals' replacement by modern humans highlight the role of environmental pressures or attribute the Neanderthals' demise to competition with modern humans, who occupied the same ecological niche. The latter assume that modern humans benefited from some selective advantage over Neanderthals, which led to the their extinction. Here we show that a scenario of migration and selectively neutral species drift predicts the Neanderthals' replacement. Our model offers a parsimonious alternative to those that invoke external factors or selective advantage, and represents a null hypothesis for assessing such alternatives. For a wide range of parameters, this hypothesis cannot be rejected. Moreover, we suggest that although selection and environmental factors may or may not have played a role in the inter-species dynamics of Neanderthals and modern humans, the eventual replacement of the Neanderthals was determined by the repeated migration of modern humans from Africa into Eurasia.

---

[1] Department of Biology, Stanford University, Stanford, CA 94305, USA. Correspondence and requests for materials should be addressed to O.K. (email: okolodny@stanford.edu)

One of the most intriguing questions concerning the evolution of modern humans is their relationship with other hominid species, particularly in light of recent findings showing that the genomes of modern humans carry the traces of introgression events with Neanderthals and Denisovans[1–5]. Although many details of the process remain unclear, archaeological and genetic evidence suggests that near the end of the middle Paleolithic, modern humans (henceforth "Moderns") migrated out of Africa, where they had evolved and where their population was large, into the Levant and thence to other parts of Eurasia[6–12]. As migrating bands of Moderns expanded the species' range, they encountered small populations of other hominid species—Neanderthals, Denisovans, and perhaps others—that seem to have occupied an ecological niche very similar to their own[13–16] (see also ref. [17] and following commentaries). Archaeological findings point to a period of 10,000–15,000 years during which Moderns and Neanderthals coexisted in the Levant and Europe, including a few thousand years in western Europe in conjunction with some regional overlap, and possibly also including recurrent replacement of one species by another in particular dwelling sites[18–28] (see Supplementary Note [1] for discussion and consideration of a shorter period of overlap. "Coexistence" refers to contemporaneous habitation of parts of Europe and the Levant, not necessarily in fully overlapping regions, see Supplemenetary Notes [3] and [4]). The two species' temporary coexistence ended in the complete disappearance of Neanderthals by 38,000 years BP (ref. [29]; data regarding Denisovans are scarce, and not discussed here). A recent analysis of ancient DNA from an eastern Neanderthal suggests that introgression of Moderns into Neanderthal populations had occurred much earlier, roughly 100,000 BP; i.e., the archaeologically established period of overlap seems to have been preceded by earlier encounters between the two species[30, 31]. This should not come as a surprise: Moderns' remains are found in the Levant as early as 120,000 BP, and the evidence suggests plausible contemporaneous overlap between the two species' ranges in the Levant for tens of thousands of years, prior to the Moderns' expansion into Europe[17, 27, 28].

Hypotheses regarding the causes of Neanderthal replacement fall into two broad, not mutually exclusive, categories. The first highlights environmental factors, such as climate change and epidemics, as the causative agents. The second attributes the Neanderthals' replacement to direct or indirect competition with Moderns, in which Moderns had some selective advantage, possibly due to a wider dietary breadth, a more efficient mode of subsistence, advantageous differences in life history, or—most prominently—a superior cognitive capacity, potentially reflected in material culture and tool use, symbolic thought as supported by artistic expression, and language. A recent study has shown that even cultural differences alone, potentially interacting with population size differences, could have provided Moderns with a critical selective edge. See detailed references in Supplementary Table 1.

Many studies that assign a major role to a selective advantage of Moderns in the Neanderthals' demise do so based on the premise that such an advantage had to exist in order to explain the Neanderthal's demise, and they focus on determining what the selective advantage could have been. In this study we show that this assumption is unnecessary: selection may have played a role in the Neanderthals' replacement, but the replacement could also have been the result of selectively neutral demographic processes, a parsimonious alternative that should a priori be preferred. Our simple model suggests that recurring migration from Africa into the Levant and Europe—even at a low rate—was sufficient to result in the Neanderthals' replacement even if neither species had a selective advantage over the other, and

regardless of possible differences in population size between the two species. This replacement is found to have been extremely likely even if migration were bidirectional, when the estimated demographic state of affairs at the time is taken into account: a small Neanderthal population in Europe and the Levant, and a larger Modern population in Africa (see also supplementary note 3). We model Moderns and Neanderthals, for simplicity, as two non-interbreeding populations that initially occupy two separate demes, respectively, (1) Africa, and (2) Europe and the Levant (henceforth "Europe"). We assume no selective differences between these populations, and simulate a neutral drift process: in every time step, a band of hunter-gatherers stochastically dies out, representing a local extinction event, and is replaced by a replicate of a band (a "propagule") found in the same deme, chosen at random, irrespective of its species' identity. In the main condition we consider, propagules of bands from Africa may migrate to Europe at a low rate; later, we also consider bidirectional migration. If a migrating propagule happens to be chosen to replace a band that died out on that time step, it "establishes". Otherwise, it dies out. Each simulation is continued until one species goes extinct. In addition, we explore a second, spatially explicit, version of the model, in which replacement of a band that died out may occur only by a propagule of a neighboring band. See Methods for more details.

Our model plays two roles in the study of the relations between Moderns and Neanderthals. First, it acts as a null model, a parsimonious alternative to models of replacement that invoke selective advantage or environmental factors to explain the replacement. Second, appreciation of the fact that Neanderthals are expected to have been replaced by Moderns regardless of any possible selective advantage to the latter is in itself paramount to attempts to reconstruct hominin evolution. That is, our finding would be important as a baseline for understanding Neanderthal–Modern dynamics, even if there were clear evidence that selection did play a role in the replacement process.

## Results

**Unidirectional migration leads to Neanderthal exclusion.** We suppose that all individuals in deme 2 (Africa) are Moderns, and $M_2 > 0$, $M_1 = 0$ (migration occurs only from deme 2 to deme 1, Europe). This results in complete replacement of Neanderthals by Moderns, regardless of the size of $M_2$ or the relative values of $N_1$, $N_2$, the carrying capacities of the two demes (see Methods for definition of all parameters and variables). This is because there is constant influx of Moderns into Europe, while within Europe stochastic drift takes place. Thus the process can be viewed as a random walk with a single absorbing boundary: if the frequency of Neanderthal bands in Europe reaches zero, there is no further change, while zero Modern bands in Europe is not an absorbing state due to continued migration from Africa.

Migration and carrying capacity in unidirectional migration: Numerical stochastic simulations of the process described by Eqs. (1)–(3) (see Methods) reveal a number of interesting aspects of the process of species replacement; for example, the relationship of the hominid band carrying capacity in Europe, $N_1$, to the time scale on which the replacement occurs and to the number of successful establishment events of migrating propagules. For a fixed probability of migration, $M_2$, we find that as Europe's carrying capacity, $N_1$, becomes larger, proportionally more migrations are required before one of the propagules from Africa establishes and ultimately leads to species' replacement (Fig. 1a). The time that it takes for successful establishment and subsequent fixation to occur is nearly proportional to $(N_1)^2$ (Fig. 1b), because the probability that a migrating propagule will establish is proportional to $1/N_1$ and the mean time from successful establishment to fixation is proportional to $N_1$ (see ref. [32]). See

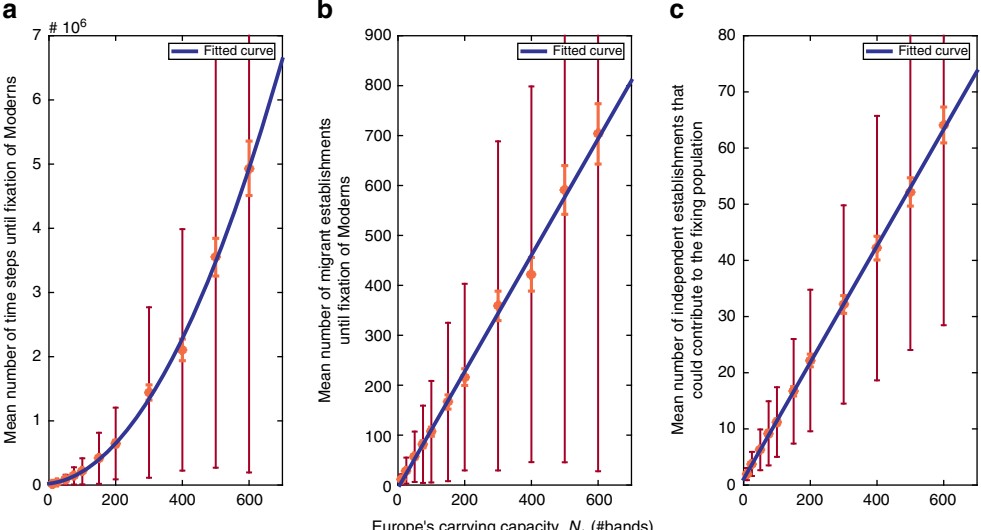

**Fig. 1** The time and number of successful migrations until species replacement. **a** Mean number of simulation time steps until complete replacement of Neanderthals by Moderns. Here and in (**b**) and (**c**), orange error bars denote two standard errors around the mean, and brown bars denote the standard deviation; 500 replicate simulation runs were conducted with each carrying capacity value. The dark blue line is a fit of the means to a quadratic function, demonstrating that the number of time steps to fixation scales with $(N_1)^2$. **b** The mean number of migrant establishments in Europe that take place until replacement occurs is linearly related to $N_1$. Here and in (**c**) the dark blue line is a fit to a linear function. **c** The number of migrant establishments that occur while the Moderns are segregating in Europe on their way to fixation scales with $N_1$. For all panels, the probability of migration into Europe per time step, $M_2 = 0.1$ ($N_2 = 100$, $m_2 = 0.001$)

Supplementary Note 2 for analogous results with different migration rates.

Another result concerns the effects of the rate of migration and the European carrying capacity, $N_1$, on the mean number of migration events into Europe that may contribute to the Modern population at the time of fixation; that is, how many migrant propagules might eventually have offspring in the population? When migration is rare ($M_2$ is small) or $N_1$ is small, a single migrating propagule may establish and drift to fixation without any subsequent Modern establishment events taking place during the process. When migration is sufficiently large, it is likely that more than one establishment event occurs before fixation of Moderns, and each of these migrations may contribute to the eventual composition of the population of Moderns in Europe (Fig. 1c). To demonstrate this we kept the migration rate constant and ran stochastic simulations of Eqs. (1)–(3) with different population sizes $N_1$ (keeping population size constant and altering migration rate gives similar results). Figure 1c shows that the number of potential contributors to the fixing population rises proportionally to the carrying capacity, $N_1$. This is because the mean time that Moderns segregate in the population until fixation scales with the population size. This pattern bridges what may seem to be a gap between our model's assumptions of ongoing migration from Africa to Europe, and suggests that the archaeological record does not support more than a single out-of-Africa event into the Levant and from it to Europe: evidence of a single migration event is to be expected under our model if the rate of migration is low or if Europe's carrying capacity is small (Fig. 1c).

These relationships may be useful in testing our model with empirical evidence concerning the replacement process, because such relationships should leave signatures in the archaeological and genetic records.

Comparing replacement duration to empirical data: Fig. 2a presents the ranges of durations of species' coexistence for various values for $N_1$, the carrying capacity in Europe, over 500 simulations per value of $N_1$. These ranges are split into the 5% fastest

replacements (orange) and the remaining 95% (dark blue), thus providing the information about the distribution of replacement durations in our neutral model, which is sufficient to conduct a statistical test to check whether our model, which forms a null hypothesis, should be rejected. The results are presented in units of the average number of times that each territory that supports a band of hominids "changed hands" during the replacement process. This unit is used for ease of interpretation, although the model is not spatially explicit, and is calculated as the overall number of events in which a band died out and was replaced by another (of the same or of the other species), divided by the population size in units of number of bands, $N_1$. The results are derived for a single rate of migration, of a migrant propagule arriving in Europe every 10 time steps on average, i.e., $M_2 = 0.1$. Supplementary Figs 1 and 2 compare the mean and median replacement durations for different rates of migration, and five possible methods for measuring the time period (Supplementary Note 1). Most methods yield similar results over a wide range of migration rates.

Figure 2b and Supplementary Fig. 3b show the result of a hypothesis rejection test for each combination of parameter values; both rely on the model's numerical results in Fig. 2a. The color of each point $(X_1, Y_1)$ in this panel indicates whether the model should or should not be rejected at a the $p = 0.05$ significance level for the combination of Europe's carrying capacity described by $X_1$, and the rate of band replacements (equivalent to the rate at which territories change hands) described by $Y_1$. This test is carried out using 12,000 years as the archaeologically supported period of species' coexistence (Fig. 2a), while Supplementary Fig. 3 provides analogous results under the assumption of 5000 years of species' overlap. Supplementary Note 1 includes further discussion of estimates of the duration of the species' coexistence, and analysis of the number of band replacements that occur between bands of different species.

Interpretation of Fig. 2b depends on the assumed band size and Europe's carrying capacity. For example, for a carrying capacity of

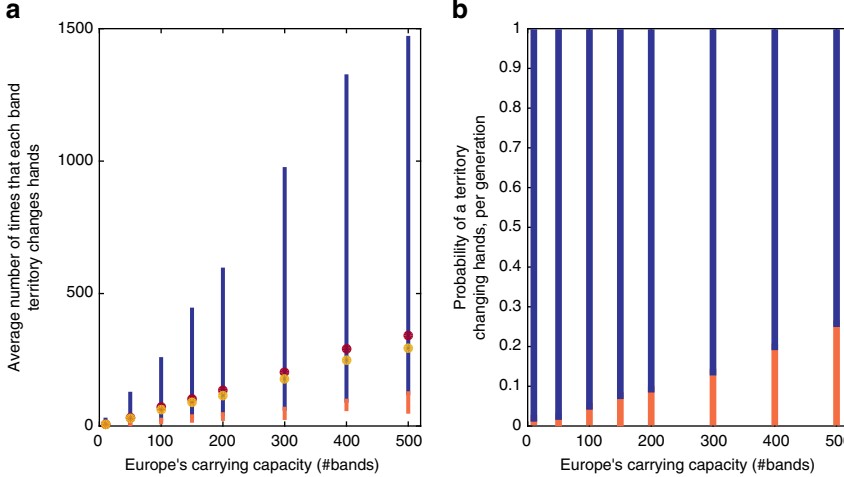

**Fig. 2** Comparing model predictions to the archaeological record. **a** Range of durations of species' coexistence for various carrying capacities in Europe, $N_1$, in units of average number of events of band replacement per band territory. The range that covers 95% of the results is marked in dark blue, and the 5% of the simulations with the shortest coexistence durations is in orange (500 simulation replicates were conducted for each value of $N_1$). The mean and median values are marked in brown and dark yellow circles, respectively. Coexistence duration is defined as the period during which both species exist in the population at frequencies above 10%, between the last crossings of this threshold by each of the species, as shown by the orange and dark yellow circles in the time trajectories of Fig. 3. See Supplementary Note 1 for more details and discussion of alternative definitions. **b** Tests of the hypothesis of neutral replacement for a range of parameter combinations, assuming a species coexistence duration of 12,000 years: each point in the panel represents the result of a test at the $p = 0.05$ significance level. The range of parameters for which neutral replacement should be rejected is denoted in orange. The range for which the model should not be rejected is marked in dark blue

5000 individuals, a slightly higher estimate than the mean population size in Eurasia suggested by Bocquet-Appel et al.[33] during this period, with a band size of 50 individuals (see refs. [34–36]), our model should be rejected only under the assumption that the rate at which band territories change hands corresponds to a probability of replacement per generation of 0.05 or lower; that is, if a territory changes hands on average less than once in every ~500 years. For a carrying capacity of 50,000 individuals (the order of magnitude according to figures suggested by Hassan[37], applied to the known habitation region in Europe in the beginning of the upper Paleolithic) but band size of 500 (in line with the definition of a tribal group in ref. [38], following ref. [39]), the rejection threshold remains the same. Rates of replacement for contemporary traditional populations summarized by Soltis et al[40]. are between 2 and 30%, but refer to group sizes on the order of many hundreds or thousands of individuals, who are sedentary and rely to a great extent on farming of crops and livestock. Bocquet-Appel et al[15] review numerous population size estimates for the Neanderthal population, and conclude that it was in the range of 5,000–70,000 individuals. Even if one adopts the high end of this range, 70,000 individuals in Eurasia, our null model should be rejected only for the lowest extreme of the range of replacement rates suggested by Soltis et al[40]. when considering—in accord with the accounts reported there—a group (band) size of 1000 individuals.

In sum, our main analysis suggests that the time scale on which species replacement took place according to the archaeological record is well within the range predicted by our model. Moreover, this null model should be rejected only if one assumes a rate of neutral band replacement or an overall hominid carrying capacity in Europe that are extreme according to ranges that have been suggested for these two variables.

The time trajectory of replacement in unidirectional migration: A potential argument against the sufficiency of a neutral model to explain Neanderthal replacement is that the archaeological evidence within continental Europe seems to point to a clear process of directional selection in which Moderns increase in frequency while Neanderthals disappear, a pattern that might not

seem to be in line with a drift explanation. This interpretation may be contested in light of recent re-assessment of archaeological finds that were initially assumed to be associated with Moderns based on their cultural complexity[41–44]. Further, even if one accepts that the process was directional, two properties of the demographic processes should be considered: one is that, although neutral, the process we describe is biased by unidirectional migration, which may underlie an increase over time in the Moderns' frequency. This may be unimportant if the probability of migration is low. A second, somewhat less intuitive consideration, is that the trajectory we should expect to have left its mark in the archaeological record according to our model is drawn from a distribution different from the one we are used to attributing to random drift; it is conditional on having reached the point of one species' fixation. A significant part of such a distribution is composed of trajectories that seem directional, particularly, as seen in Fig. 3, in the period approaching final fixation.

**Bidirectional migration between Africa and Europe.** Archaeological evidence suggests that the Levant was the southernmost tip of the Neanderthal population, and thus does not support migration of Neanderthals into Africa. However, it is possible that such migrations occurred, and we explore a number of scenarios for this bidirectional migration.

Because the two populations in our model are finite, all scenarios—apart from the case in which there is no migration between the two demes—inevitably end in fixation of one species and extinction of the other. Thus the question is how the probability of each species' fixation depends on the migration parameters and the carrying capacities of the two demes.

The initial condition in all our scenarios is that deme 1 (Europe) is populated by Neanderthals and deme 2 (Africa) by Moderns. We treat only cases in which the population size in deme 2 is equal to or greater than the population size in deme 1, i.e., $N_2 \geq N_1$. This assumption is supported by population size estimates derived from genetic data, archaeological findings, and environmental factors (Supplementary Note 3). However, some

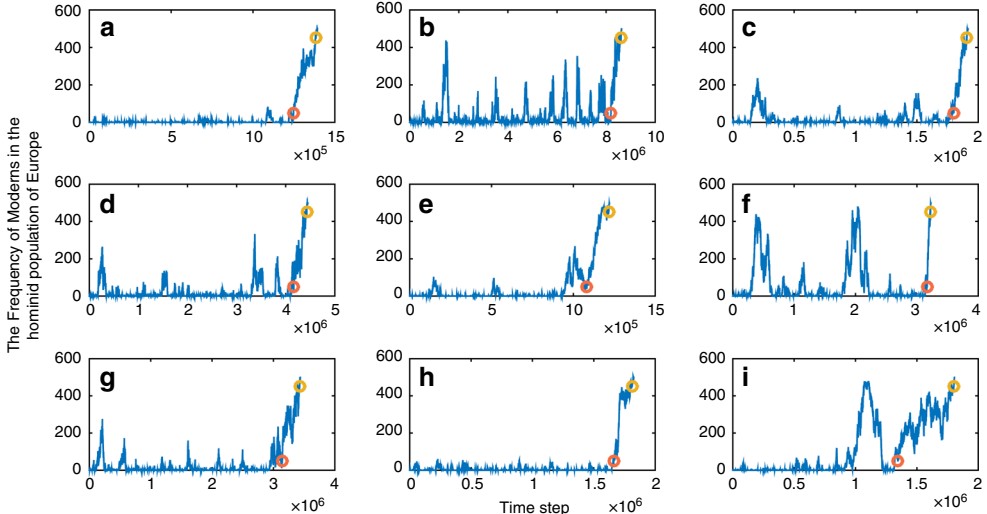

**Fig. 3** Time trajectories of separate simulation runs. Number of bands of Moderns in Europe over time in nine randomly selected simulation runs **a–i**, with $N_1 = 500$ bands in Europe, and migration probability from Africa to Europe, $M_2$, of 0.1 per time step (migration is unidirectional; $M_1 = 0$). Orange and dark yellow circles denote, respectively, the last time that the Moderns and that the Neanderthals made up at least 10% each of the hominid population in Europe. Many trajectories, particularly in the final phase leading to Modern's fixation, are highly directional

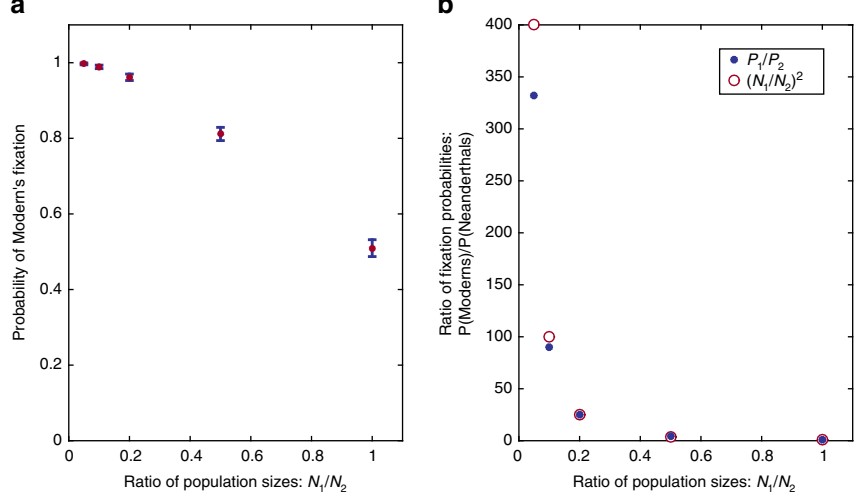

**Fig. 4** Fixation probabilities under bidirectional symmetric migration. **a** Probability of Modern's fixation (brown; blue bars indicate two standard errors around the observed frequency of fixation). **b** Ratio between Moderns' and Neanderthals' probabilities of fixation, which is approximated by the square of the inverse of the ratio of population sizes (brown) for much of the range of this ratio. It is slightly lower for scenarios in which the Moderns' population is much larger than the Neanderthals'. For all runs, $m_1 = m_2 = 0.0001$, $N_2 = 500$ (Africa), $N_1$ refers to Europe. 500 simulation replicates were conducted for each value of $N_1/N_2$

accounts of the archaeological record suggest that populations in much of Africa near the transition from the middle to upper Paleolithic were small (e.g., ref. [45]). Perhaps contemporaneous bottlenecks and reductions in the Neanderthal population size, potentially driven by glaciation, can consolidate these accounts.

Symmetric migration: If the parameters of outgoing migration are equal, i.e., $m_1 = m_2$, one might expect that the relative probabilities of fixation of the two species would be equal to the ratio of population sizes in the two demes. Somewhat non-intuitively, this is not the case; the species that is initially in the larger of the two demes (deme 2, Africa) has a fixation probability in Europe that is greater than its relative initial population size would suggest. This is because the initial conditions increase this species' probability of early success in a number of ways. First, to a good approximation, once established, progeny of a migrant from deme $x$ will drift to fixation with probability inversely proportional to $N_y$. The probability ratio of this occurrence is $N_x/N_y$ for a migrant from $x$ to fix compared to that of a migrant from $y$. Second, the number of migrants is biased in favor of outgoing migration from the larger deme: we defined the probability of migration in our model as dependent on both the migration parameter and on population size, and so migration when $m_1 = m_2$ occurs more frequently from the larger deme. Third, a migrant from deme $x$ has a probability proportional to $1/N_y$ of establishing in deme $y$ following its arrival, and thus the probability of establishment following arrival of a migrant from deme $x$ is more likely by a factor of $N_x/N_y$ than the establishment probability in deme $x$ of a migrant from deme $y$. This potential advantage is corrected for in our simulation by our definition of a time step: at each time step a band in one of the demes dies, and since the choice of band is random, more time steps are realized as dying events in the larger

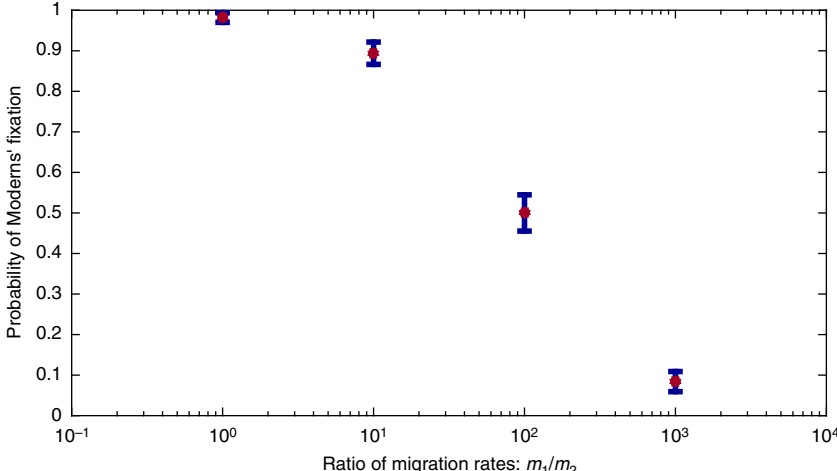

**Fig. 5** Under bidirectional migration, fixation probability depends on migration rates. The probability of Moderns' replacement of the Neanderthals for a range of ratios between the migration parameters, $m_1/m_2$, with a constant ratio of population sizes (brown; blue bars indicate two standard errors around the observed frequency of fixation). For all runs, $N_1 = 50$, $N_2 = 500$, $m_2 = 5 \times 10^{-6}$. 500 simulation replicates were conducted for each value of $m_1/m_2$

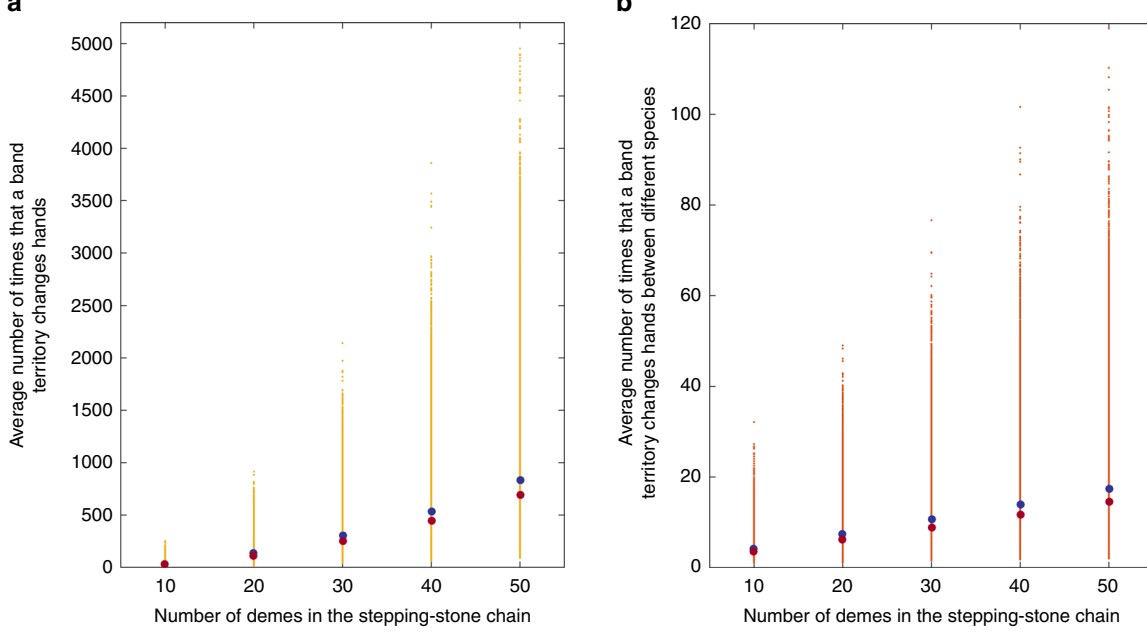

**Fig. 6** Band replacements in the spatially explicit model. The mean (dark blue) and median (brown) of the mean number of times that a band territory changes hands during the species replacement process. Panel **a** depicts these values for both band replacements that occur between bands of the same species and bands that are not of the same species. Panel **b** depicts these values for inter-species replacements only. 50,000 simulations were conducted for each value of the number of demes in the stepping-stone chain

deme; thus migrants from the smaller deme have a smaller probability of establishing, but a proportionally larger number of time steps in which such establishments may occur.

Put together, these effects suggest that the ratio of fixation probabilities, $P_1/P_2$, should reflect an advantage to the species from the larger deme that is proportional to the ratio of the population sizes to the power of two. However, these intuitions are exact only when the initial conditions have not changed, and each deme is still composed of a single species. The bias in favor of the species that originated in the larger deme is somewhat attenuated through time, and we should thus expect a ratio of fixation probabilities that reflects an advantage that is somewhat

smaller than $(N_x/N_y)^2$ to the species that originates in the larger deme. Figure 4 shows the probability of the fixation of Moderns as a function of the ratio between the two demes' population sizes.

Asymmetric migration: If $m_2 > m_1$, i.e., the parameter of migration out of Africa is larger, we find that the probability of Moderns' eventual fixation is greater than that of Neanderthals, as is expected since $N_2 > N_1$. This holds true even for some cases in which $m_2 < m_1$, as a result of the effects of the larger population size. Although unrealistic given the archaeological findings, study of various ratios between the two migration parameters, in which the parameter of migration from Europe into Africa is greater

 

than its counterpart, i.e., $m_1 > m_2$, is of interest as it allows us to explore the effects of population size differences (Fig. 5). For a population size ratio of 1:10 between Europe and Africa, we find —in line with the previous analysis—that even with a 10:1 ratio of migration rates, which creates equal probabilities of migration in the two directions, i.e., $M_1 = M_2$, the probability of Neanderthal fixation remains very low (~ 0.1). Only when the migration rate out of Europe ($m_1$) is greater by a factor of 100 than its counterpart ($m_2$) do Neanderthals and Moderns have the same probability of fixation (Fig. 5).

**The spatially explicit model**. Using the spatially explicit version of our model, we also conducted simulations for carrying capacities of 100 to 500 bands, leading to stepping-stone chains of length 10 to 50 (see Methods). We find that the mean number of times that each band territory changed hands between different bands (considering replacements both within and between species, Fig. 6a) ranges from values lower than 10 in some of the simulation runs with a carrying capacity of 100 bands, to values on the order of a few thousands in some of the simulations for carrying capacities of 400 and 500. The median for the mean number of times that each territory changed hands was—for carrying capacities 100, 200, 300, 400, and 500, respectively—28, 110, 250, 442, 694. These results are spread across a wider range than those found in the non-spatial model, but are on the same order of magnitude. The vast majority of these band replacements occur between bands of the same species; as can be seen in Fig. 6b, the mean number of times that each band territory changes hands between bands of different species is much smaller, ranging from a median of 3.5 for carrying capacity of 100 bands to 14.5 for a carrying capacity of 500 bands (for comparison with the non-spatial model, see "The number of times that each band territory switched hands between species" in Supplementary Note 1). These inter-species replacements thus constitute only a small fraction of the overall number of band replacements, ranging from circa 12% among the replacements for a carrying capacity of 100 bands, to only 2% for a carrying capacity of 500 bands. Also of note is that these are median values; there are many simulation instances in which the number of inter-species replacements per site is much lower, sometimes on the order of <2 inter-species replacements per site. This is true even for large carrying capacities, in which the overall species' replacement is on average longer.

## Discussion

We have shown that a simple selectively neutral model of population dynamics, random drift in finite populations with migration, can account for the replacement of the Neanderthals by Moderns that occurred near the transition between the middle and upper Paleolithic. Although a stochastic process, this replacement was certain to occur, even in a selectively neutral setting, given the estimated migration pattern near the onset of the interaction between the two populations, namely repeated migration of small propagules of Moderns out of Africa into the Levant and Europe. Replacement of the Neanderthals was certain to occur even for very low migration rates, as long as migration was unidirectional, regardless of the ratio between the population sizes in the two demes. In other words, the scenario that our model proposes is not one of "population swamping", but of gradual replacement, in which small individual bands of Moderns migrate out of Africa, establish in Europe, and stochastically increase in frequency until ultimately the lineage of one or a few such bands reaches fixation. We have also demonstrated that even if bidirectional migration between Europe and Africa had occurred, Moderns would have been extremely likely to

eventually replace Neanderthals, given the estimated differences in population size between the species, in favor of Moderns. This elevated likelihood of replacement stems from the disproportionate impact of initial population size differences on the probability of eventual fixation.

Further, we have realized a spatially explicit version of our model, which—although an extreme simplification of the spatial aspect of the inter-species dynamics—captures some of their qualitative features. This model's primary goal was to explore whether, or in which respects, a model that has an explicit spatial component would diverge in its results from those of the non-spatial model. Reassuringly, we find that in the overall dynamics that it portrays, and particularly with regard to the time scale on which replacement is expected to take place, this model gives results that are on the same order of magnitude as those of the non-spatial model. The aspect in which the results of the spatially explicit model are dramatically different from those of the non-spatial model is the number of inter-species band replacement events per band territory: this number is much smaller in the spatially explicit model. This finding stems from the fact that in this model, each band is surrounded by bands of the same species during most of the dynamics: each band is found most of the time away from the front of interaction between the two species. Thus, when a band dies out, it is typically replaced by a band of its own species. Such assortment of the spatial distribution of the bands is probably a more extreme case than the one that characterized the actual replacement process, while full panmixia across all of Europe, as in the non-spatial model, is at the other extreme. Further exploration, as well as developments in the analysis of archaeological findings and the excavation of more sites, may enable us in the future to assess where reality lay along this spectrum.

Our model is a parsimonious alternative to a model in which selection is the major driver in the replacement of the Neanderthals. We show that the time scale on which the replacement occurred according to the archaeological record is within the range of replacement durations predicted by our model for a wide range of parameter values, unless fairly extreme values for demographic parameters, such as Europe's carrying capacity, are assumed. As demonstrated in Supplementary Note 1, many alternative measures for the duration of the process would yield results of the same order of magnitude. It is reasonable that, under some demographic parameters beyond the range that we studied, our model would produce results that are incompatible with empirical evidence. However, the difference would not be by orders of magnitude, suggesting that a model of selectively neutral replacement should be considered under a very wide range of scenarios.

In addition to the duration of the replacement process, it would be desirable to discern other patterns that might distinguish a scenario of neutral species' replacement from one that is driven by selection. The characteristics of the Moderns' frequency trajectory over time may provide such a pattern: although many trajectories of neutral fixation seem very directional and do not differ significantly from the near-deterministic trajectory expected under a selection scenario, some trajectories are far from monotonic (e.g., Fig. 3i) in their eventual ascent towards fixation, and many are characterized by early phases that seem disjoint from the final phases of the fixation process. In these early phases, the Moderns' frequency reaches intermediate values and then decreases (e.g., Fig. 3b, c, d, f, g). Such a pattern is not expected under a selective scenario, which is predicted to produce a near-deterministic trajectory once some minimal frequency threshold is crossed[32]. In other words, an increase of Moderns to intermediate frequency, followed by a significant drop, is expected to occur in some, but not in all, cases of neutral replacement, and is extremely unlikely under a selection-driven replacement process.

If evidence of a transient Moderns' intermediate frequency were found, it would argue against the role of (strong) selection in the replacement process.

Such early increases to intermediate frequency are likely to go undetected in the archaeological record for many reasons—its sparseness, the low likelihood of uncovering skeletal remains in a site populated for a short time, and the disconnect from a long-term archaeological context that would help to shed light on the species' identity. Also, early Moderns in Europe were characterized by material cultures different from those associated with the species during their later, well-established, period of existence there, which may increase the likelihood that the remains would be misclassified as Neanderthal (see also "Calculation of the duration of replacement" in Supplementary Notes 1 and 4); indeed, a recent detailed analysis of lithic technologies suggests that such misclassification may have occurred, and supports a model of multiple early Modern migrations out of Africa that reached intermediate frequencies in Eurasia[46]. Similarly, findings of anatomically modern humans in the Levant in multiple sites from the late middle Paleolithic suggest a probable species' overlap in this region in considerable numbers over potentially long durations[27]. Recent developments in the field of ancient genomics may shed light on this question by providing evidence of early introgression events[30, 31]. The probability of detectable successful introgression as a result of any single contact between two species is low, with reasons ranging from the low likelihood of the occurrence of productive sexual contact in the first place, to potential hybrid fitness disadvantage, and drift and possible selection acting to obscure or eliminate the genomic traces of such introgression[47–50]. Thus, observed introgression is likely to be a reflection of a history of substantial inter-species contact, which is most likely to have occurred when both species were in the same region at intermediate frequencies. Evidence of at least one early introgression from Moderns into a subgroup of Neanderthals has recently been found[31], and is estimated to have occurred well before Moderns are identified in the European archaeological record. This lends support—according to the prediction in the previous paragraph—to a re-evaluation of the time trajectory of Moderns in Europe and the Levant that is not in line with their having a (strong) selective advantage.

Effectively neutral replacement could also have occurred under a fairly broad range of conditions, in which selection acts on differences between the two species: under a range of conditions, efficiency of selection is approximately proportional to population size[32]. Because of the relatively small population sizes of both species, even if differences between the species had given one of them a selective advantage, this advantage may have had little effect at the population level, leaving the system in a nearly neutral regime for which our model would hold quite well. This would have been the case for a fitness advantage $s$ such that $N \cdot s$ is near 1, i.e., for an overall fitness advantage of $s = 10^{-4}$ or lower. Because the initial interaction between the groups was probably more localized than in our model and probably occurred between small subgroups of the two populations, the range of selection coefficients for which the species' interaction would have been within the nearly neutral regime, at least during part of the demographic process, is realistically even broader, perhaps up to an advantage of $s = 10^{-3}$ or $s = 10^{-2}$ to one of the species, if the sizes of the interacting populations were on the order of hundreds or thousands of individuals. In other words, even if Neanderthals had a selective advantage, unless it was very large, they are likely to have been eventually replaced by species drift. Introgression between the two species might have mitigated this effect, potentially allowing one or both species to incorporate advantageous alleles from the other species and to radically reduce the selective

differences between the two species, even if admixture was limited and even if hybrid lineages initially had some selective disadvantage (see references in Supplementary Table 2).

We do not suggest that there were no differences between the species that had an effect on fitness; on the contrary: morphological and genetic differences between the two species suggest that they differed phenotypically in ways that are highly likely to have affected fitness. Which of these differences conferred a selective advantage is debatable, and how such selective differences in various traits acted jointly to affect the overall fitness will likely remain unknown. Arguments in favor of a selective advantage to either species with regard to sets of traits are compelling, ranging from potential advantage to the Neanderthals stemming from adaptation to local conditions such as climate and pathogens[9, 51], to selective advantage of Moderns due merely to their greater overall population size and proportionally smaller predicted mutational load[49], to differences in morphology, mode of subsistence, cultural aspects such as tool use, and, possibly, cognitive capacity (see supplementary Table 1). Such studies are important for attempts to unravel the evolutionary history of the two species, but their interpretation in this context must be cautious. Besides providing support for probable selective advantages to each species over its counterpart with regard to particular traits, and thus not providing a decisive conclusion regarding the overall species' relative fitnesses, many of these studies compare material findings associated with the two species in sites that are not contemporaneous (see, e.g., refs. [52, 53]). Such a comparison in order to reconstruct the dynamics of species' replacement should be made with great caution[54]. Our study demonstrates that species replacement would be expected under a neutral model, in a manner compatible with the replacement that actually took place, and that neutral processes are able to account for the inter-species interaction regardless of whether selective differences between the two species existed.

An extensive review of the arguments related to the possible selective advantage of Neanderthals over Moderns or vice versa is beyond the scope of this study. We find it important to address one major line of argument in this context, which suggests that Moderns had a cognitive and cultural advantage, potentially in the form of symbolic thought or language, over Neanderthals (see refs. [9, 45], and those in Supplementary Table 1). To date, genetic and cranio-morphological comparisons between the species have not produced any unequivocal evidence that would support this argument, which is grounded mostly in the material archaeological record of artifacts and cave drawings that seemingly provide fairly convincing circumstantial evidence: during the period of co-habitation in Europe of the two species and within the first ten millennia that followed it, a demographic and cultural revolution occurred in Europe[55]. Population densities increased by a factor of 2–10 in many localities[56], previously uninhabited regions were colonized[57, 58], forms of artistic expression became much more common than before[55], and the repertoire and complexity of tools grew dramatically[45, 59, 60].

Whether Neanderthals were responsible for some of these novelties and whether the revolution was as sudden as initially thought has attracted much discussion[42, 43, 61–68], as has the possibility that Neanderthals and Moderns had significant cultural exchange, suggesting that they were—at the least—comparable, if not on par, in their cognitive abilities[60, 69].

We suggest a number of additional arguments that call for a guarded interpretation of the demographic and cultural shifts as reflecting a selective advantage of Moderns. Undoubtedly, the living conditions of both species changed extensively during this period of time: Moderns' migration to new localities exposed them to novel challenges, and both species were faced with increased hominin competition and exposure to new ideas and

practices together with—potentially—direct competition different from any that preceded it. These may have been accompanied or preceded by significant independent environmental changes[70–74]. In light of these changes, extensive demographic and/or cultural changes were likely to have occurred in both species[67, 75], even if the species did not differ in their cognitive capacities and even if no change in cognitive abilities occurred throughout this period (suggesting speculation of such differences, e.g., refs. [9, 45], is unwarranted). Dramatic punctuated changes occur in many biological systems[76, 77], and are particularly likely to be triggered by extensive changes such as the ones that the Neanderthals and Moderns went through. In previous work we have demonstrated that such sudden change is specifically to be expected in the evolution of culture[78], especially upon exposure to cultural novelties, which can easily trigger innovations by analogy or by combination with existing practices. Such cultural changes can lead to a further rapid, possibly exponential, rise in cultural complexity[78–81], which may in turn prompt demographic change. In other words, we suggest that the increase in cultural complexity that is found in Europe near the replacement of Neanderthals by Moderns may be the result of the Moderns' geographic expansion and of the two species' interaction, rather than the cause of the replacement or its driver (see also ref. [28]).

A second relevant observation is that if a cognitive and cultural advantage were a driver of the Moderns' spread from Africa into the Levant and from there to Europe, one might expect to find cultural continuity between archaeological sites along this route near the transition from the middle to the upper Paleolithic. As has been pointed out and widely discussed, there is no clear-cut evidence for such continuity (see, e.g., refs. [20, 28, 55, 67, 75]). Moreover, the material cultures associated with Moderns and Neanderthals in the Levant during the late middle Paleolithic, in the period preceding the replacement, are indistinguishable from one another (see, e.g., refs. [27, 28]). The appearance of advanced cultural features in Europe and the Levant only after species' interaction was likely to have taken place is in line with our suggestion that advanced culture is an outcome of this interaction (and see, e.g., refs. [64, 82]). Similarly, if the cultural burst is a result of the species' interaction, one may expect to find in the emerging cultures some characteristics of the individual cultures that preceded them, including continuity of some local features in some regions. This may be the case in variants of the Ahmarian, Aurignacian and transitional techno-complexes from this period (see, e.g., ref. [58] and those in Supplementary Table 2). Finally, the "full package" of upper Paleolithic modernity appears in most of the regions that were populated after the Neanderthals' replacement—Siberia, East Asia, and the Sahul—only between 10,000–20,000 years later, suggesting it may have developed only in particular populations after the replacement had occurred or as it was taking place, and that its role in the replacement and the moderns' geographic spread was limited; see references in Supplementary Table 2).

In sum, we do not endorse any particular stance as to whether Moderns did or did not have a cognitive or cultural advantage over Neanderthals, but point out that much of the evidence in support of this claim should be interpreted cautiously.

We suggest that migration dynamics together with events of local dispersal and replacement, in a selectively neutral model, can explain the Neanderthal–Modern interaction and subsequent replacement of the former by the latter, without invoking selection or external environmental factors, and even—under some scenarios—regardless of possible difference in population size between the species. Advanced methods of dating archaeological findings and new methods in ancient genomics are expected to provide more detailed information within the next few years; Combined with models of the two species' interaction that take into account geographical sub-units, population substructure, and introgression, new empirical data may soon enable us to further elucidate the dynamics that led to the Neanderthals' replacement and to assess in more detail whether a neutral model such as that we propose is sufficient to explain this process and its outcome.

## Methods

**A neutral model of species' replacement.** We suggest that a model of migration and neutral species drift can explain the replacement of Neanderthals by Moderns and is in line with the evidence to date. Our model assumes no selective differences between the two species; that is, the competitive interaction between individuals or groups from the different species is identical to the competitive interaction between individuals or groups within the same species. Thus the two species are equivalent to two non-interbreeding subgroups of a single species. Although the two species did interbreed to some extent[4, 30, 48], for simplicity we do not incorporate introgression into our model. In it, the only trait of interest is the species' identity of individuals or groups; this formally equates the model with a simple, well-studied scenario: two selectively neutral alleles segregating at a genetic locus[32]. Perhaps the most fundamental property of such a scenario (in the absence of mutation) is that random drift will ultimately lead to the fixation of one allele and the extinction of the other. Applied to species, the analogous process has been termed "species drift"[76].

This portrayal of the Neanderthals–Moderns situation is already sufficient to explain why one of the species had to eventually disappear, and is in line with the archaeological evidence that points to a period of co-occurrence of the two species in Europe and the Levant. However, in order to understand how and why the two species' history would necessarily result in the Neanderthals' extinction, we must take into account geographic and demographic aspects of the two species' populations at the time. To do so, we model Europe and the Levant (deme 1, for simplicity, referred to henceforth as "Europe") and Africa (deme 2, "Africa") as separate demes with migration between them. The two demes have constant but possibly different hominin carrying capacities. See Supplementary Note 3 for details.

For realism and simplicity, we consider the dynamics of bands of individuals. That is, the entities whose fate is tracked in our model are small groups of individuals: such a band may die out by chance and be replaced by a propagule from another band (similar to the "propagule pool" model described in ref. [83], see also refs. [34–36], a propagule should be regarded as a copy of its band of origin). The carrying capacity of bands that reside in deme $x$ ($x = 1,2$) is the constant $N_x$. The probability of outgoing migration of a propagule from deme $x$ per time step is denoted $M_x$, and is proportional to a parameter $m_x$ and to $N_x$. The rate of migration is assumed to be small enough that at most a single propagule can migrate per time step; accordingly, if $m_x \cdot N_x > 1$, we set $M_x = 1$, in which case migration occurs with probability 1 at every time step.

The population dynamics are those of a birth-death process akin to a Moran process with migration: at every time step a band chosen at random (regardless of its species' identity) dies out, and is randomly replaced by a propagule from one of the other bands in its deme (Africa or Europe) or by a migrant propagule that had arrived from the other deme during the most recent time step. These stochastic death-and-replacement dynamics are a result of local extinctions due to environmental fluctuations, stochasticity in reproductive success, or competition among bands. This competition, which is assumed to be identically harsh between any two bands regardless of their species' identity, may include inter-group competition for resources (e.g., territory, food, dwelling sites) and even direct inter-group violence. Importantly, the result—which of the competing bands will die out —is independent of the bands' species' identity (see also Supplementary Note 3, section 4). Note that one implication of these dynamics is that the population within each deme is fully mixed, i.e., this model does not account for the geographic spatial structure within each deme. Below we present a simple model that considers spatial structure.

We use the term "establishment" to describe the case in which a propagule migrated and was chosen to replace a band that died out. The probability of a migrant propagule's establishment in deme $x$ after arriving from deme $y$ is thus $1/N_x$. Only propagules of existing bands migrate; thus migration has no effect on the population in its deme of origin. These dynamics ensure that the "population size" in each deme is constant, equal to the "carrying capacity" of that deme, $N_x$. Except where noted otherwise, both terms refer henceforth to the number of bands in a deme.

The following set of transition probabilities constitutes a full mathematical description of this model. Let $i_x$ denote the number of bands of Moderns in deme $x$, and $P_{i_x \to i_x+1}$ denote the probability per time step, at time $t$, of an increase by 1 of $i_x$, $P_{i_x \to i_x-1}$ the probability of decrease by 1 of $i_x$, and $P_{i_x \to i_x}$ the probability of no change to $i_x$. As indicated by the lower subscript of $i_x$, Eqs. (1)–(3) describe the dynamics in deme 1, representing Europe, and equations for the dynamics in deme 2, representing Africa, are obtained by replacing every 1 by 2 and 2 by 1 in

Eqs. (1)–(3).

$$P_{i_1 \to i_1+1} = \frac{N_1 - i_1}{N_1}\left[(1 - M_2)\cdot\frac{i_1}{N_1 - 1} + M_2\cdot\left(\left(\frac{i_2}{N_2}\right)\cdot\frac{i_1+1}{N_1} + \left(\frac{N_2 - i_2}{N_2}\right)\cdot\frac{i_1}{N_1}\right)\right] \quad (1)$$

$$P_{i_1 \to i_1-1} = \frac{i_1}{N_1}\left[(1 - M_2)\cdot\frac{N_1 - i_1}{N_1 - 1} + M_2\cdot\left(\left(\frac{i_2}{N_2}\right)\cdot\frac{N_1 - i_1}{N_1} + \left(\frac{N_2 - i_2}{N_2}\right)\cdot\frac{N_1 - i_1 + 1}{N_1}\right)\right] \quad (2)$$

$$P_{i_1 \to i_1} = 1 - (P_{i_1 \to i_1+1} + P_{i_1 \to i_1-1}) \quad (3)$$

These equations are derived as follows: In Eq. 1, $\left(\frac{N_1 - i_1}{N_1}\right)$ is the probability that in the current time step, a Neanderthal band in deme 1 dies out (otherwise an increase in the Modern's population in this deme during this time step is impossible). The terms $(1 - M_2)$ and $M_2$, respectively, represent the probabilities that migration from deme 2 did not occur and that it did occur. The term $\frac{i_1}{N_1-1}$ is the probability that the propagule chosen to replace the one that died out in deme 1 (Europe) is Modern, given that no migration occurred; thus the number of candidate propagules that can act as a replacement is $N_1 - 1$. $\frac{i_2}{N_2}$ and $\frac{i_1+1}{N_1}$ represent, respectively, the probabilities that the migrant propagule to Europe is Modern, and that a Modern propagule is chosen to replace the band that died out. Another possibility that increases the Modern population in Europe, given that migration had occurred, is represented by $\left(\frac{N_2-i_2}{N_2}\right)\cdot\frac{i_1}{N_1}$, i.e., the migrant to Europe is Neanderthal, and yet a Modern propagule is chosen to replace the band that died out. Eq. 2 is composed of analogous constituents, whose interpretation is analogous to the description above. This model is similar (but not identical) to the Moran process with mutation that is studied, for example, by Ewens (ref. [32], p. 106), if one of the migration probabilities is zero. The expected number of time steps required for species' exclusion, in the process described by Eqs. (1)–(3), can be calculated using equations 2.144 and 2.160 in ref. [32], and is not provided here.

We limit the scope of our exploration to conditions in which one of the demes (deme 1, representing Europe) is initially populated only by bands of Neanderthals, and deme 2, representing Africa, is initially populated only by bands of Moderns. We first analyze the case in which migration occurs only from Africa to Europe (deme 2 to deme 1), the scenario that is widely believed to have taken place near to and during the interaction between the Neanderthal and Modern populations, based on the lack of evidence so far that would support Neanderthals' existence in Africa[6–12]. For completeness, we then report simulation results for the case in which migration occurs in both directions, with both equal and unequal migration rates in the two directions.

**Methods for comparing model results to empirical evidence**. The replacement of Neanderthals by Moderns seems to have occurred surprisingly fast when compared to archaeological and evolutionary time scales of the two species' existence. This may be why many scholars assume that the process was necessarily driven by selection, but whether the process should be considered as having been rapid depends on properties of the model assumed, such as the Neanderthal population size and the expected duration of replacement. Our model, which takes

into account major aspects of the two species' demography at the time but does not include selection, may be regarded as a baseline, or a null model, for such an evaluation. To assess whether this null model can be rejected in favor of a selection scenario, we study the distribution of replacement durations produced by the model and compare it to the replacement duration suggested by empirical evidence. Such an attempt faces a number of obstacles.

First, it is notoriously hard to correlate the time units in evolutionary models with the time span of real-life scenarios. This is also the case in our model, which entails the selectively neutral replacement of bands, whose empirical rate is hard to gauge and for which there are no clear estimates. Second, the species' replacement duration in our model depends critically on the parameter $N_1$, the number of bands in Europe. Estimates of hominid population sizes in Europe near the end of the middle Paleolithic vary over more than an order of magnitude and—according to reconstructions of paleo-climate during this era—are likely to have changed significantly while the replacement was taking place[15]. Finally, the way in which replacement duration is estimated may affect the result by more than an order of magnitude. Comparison between the simulations and the archaeologically estimated period of coexistence of the two species should take into account the time point at which species' coexistence is likely to be evident in the archaeological record. That is, the appropriate duration to be compared should not be the full duration of each model simulation, but the period during which both species are likely to have a demonstrable presence. This would be based on archaeological findings that can be clearly associated with the identity of the species that produced them, and could, for example, be the period from the initial crossing of some frequency threshold by the Moderns until the Neanderthals constitute less than this threshold in the overall European population, or the period between the last crossings of these thresholds (see Supplementary Note 1 for a discussion of alternatives).

Accordingly, we have conducted statistical analyses for a range of combinations of parameters. We attempt to assess, under a range of possible choices and assumptions, whether the time required for species replacement according to our model is within a reasonable range compared to the empirical evidence. Because our model serves as a null model relative to models of species replacement that are not neutral (for example, that assume selection), we test, for each combination of parameters, whether our model can be rejected at the $p = 0.05$ significance level. For this, we ask whether the archaeologically supported duration of species coexistence falls, for example, in or below the range of the 5% shortest coexistence durations produced by our model, in which case our model should be rejected, or outside of this range (if one wishes to conduct this test with a lower rate of false rejections, i.e., using a more stringent $p$-value, the 5% threshold should be decreased accordingly). In our analysis, the considered duration of species coexistence is the time from the last simulation time step in which Moderns comprised 10% of the population in Europe until the last time that they reached 90% of it (see Supplementary Note 1 for a discussion of this choice and

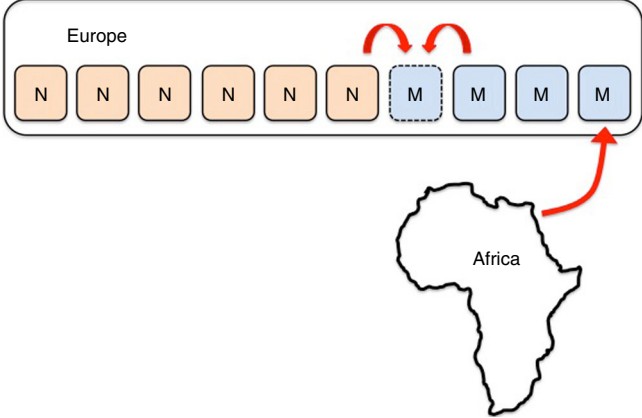

**Fig. 7** An illustration of the 1-D stepping-stone model. Each deme is occupied by a single band, whose species identity is denoted by M (Modern) or N (Neanderthal). When a band stochastically dies out, it is replaced by a propagule from one of its immediate neighbors. Migration of Moderns from Africa occurs to the rightmost deme within Europe, and is assumed to occur immediately if the band in this deme dies out

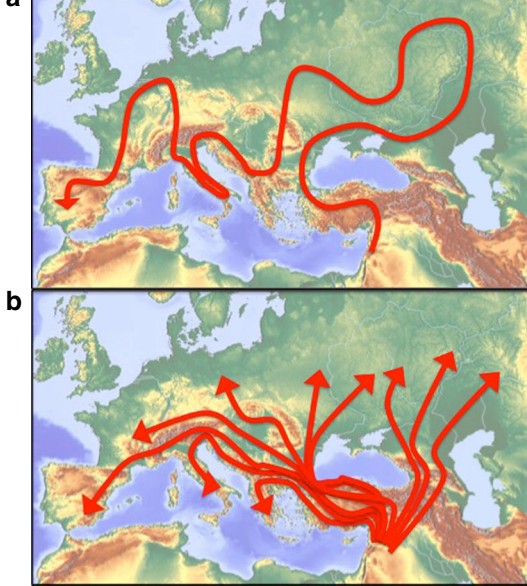

**Fig. 8** Alternative models of Moderns' spatial trajectory in Europe. Schematic illustrations of modeling alternatives that differ in the number of trajectories along which Moderns may spread into Europe. One trajectory in **a**, 10 trajectories in **b**. Each trajectory is modeled as a single 1-D series of demes. The model realizes the dynamics along a single trajectory

alternatives). The rates are described in units of the probability of band replacement per generation (25 years).

Each of our model's time steps is equal to a stochastic replacement of one band by another. The rate of band replacement, the characteristic band size, and the hominid carrying capacity of Europe in the middle Paleolithic, are variables that are not necessary in order to derive the numerical results of our model, but an estimate of these is required, post-simulation, in order to compare the results to the period of species' coexistence in the archaeological record and determine whether the model should be rejected. We conduct simulations for a broad range of carrying capacities in Europe, ranging from 10 to 500 bands. This covers the full range of estimates suggested in the literature, with the size of bands, or identity-groups, ranging from 50 to 1000[34–40] and with population sizes ranging from 5000 to 70,000[37, 33, 15]. Considering the rate of band replacements in units of the per-generation probability that a band is replaced, we determine whether our model should be rejected for the full range of possible values of this variable, with the per-generation replacement probability ranging from 0 to 1.

**A spatial model of species' replacement**. Any model, particularly one such as ours that aims to describe the null expectation, is a highly simplified version of reality. The aspect in which this simplification deviates most significantly from the reality of the inter-species dynamics is that it does not account for the spatial structure of the dynamics beyond the split into two major demes. A full treatment of the spatial complexity of the dynamics requires complex modeling and a large number of strong assumptions, making the model less general and more sensitive to specific details, over which there is no agreement among researchers. Such treatment is well beyond the scope of our current study. However, to check whether spatial structure would lead to qualitatively different results from those of the non-spatial model, we have examined a simple model that captures the effect of a spatial component in the inter-species dynamics. We designed it such that it would be easily tractable and intuitive in its structure, and would require a minimal number of assumptions.

We model the spread of Moderns into Eurasia as a stepping-stone model[84], where the inter-species dynamics play out on a shifting front of contact, i.e., by reducing the 2-D spatial structure of Eurasia (henceforth "Europe") to a 1-D series of single-band territories, as illustrated in Fig. 7, where a species range increases or decreases via replacement of the foremost band of one species by the other. As previously, we consider a Moran model, where at each time step one band stochastically dies out (chosen regardless of its species identity, reflecting the null assumption that the species are selectively equivalent) and is replaced by a randomly chosen neighboring band. As illustrated in Fig. 7, in this model, each band has only two neighboring bands.

We use this model to study only the scenario of unidirectional migration out of Africa. We focus on the duration of co-habitation of parts of Europe by bands of the two species; the period of interest always begins once both species are found within Europe, and so the migration dynamics need not be modeled explicitly. This is realized by assuming that the first (henceforth, "rightmost") stepping stone (henceforth, "deme") would be habitable only by Moderns; Neanderthals cannot establish in this deme, and if the band in it happens to die out, it is immediately replaced by a Modern band, representing a successful establishment of a migrant band from Africa. The initial conditions of each simulation run are that all other demes are initially occupied by Neanderthals.

As in the model of unidirectional migration described previously, each simulation is determined to end in full Neanderthal replacement, because the system has a single attracting state: the rightmost deme is always populated by Moderns, which may spread from there leftwards via drift but can never go extinct, while Neanderthals initially populate all other demes. If the drift dynamics gradually lead to all of the demes being populated by Moderns, the Neanderthal's population cannot be replenished, and they go extinct. The question of interest is primarily how long this process of neutral species replacement via random drift is expected to be. Because the average time between band replacements is a parameter in the analysis, which should be derived from empirical evidence and that does not affect the model's result when time is measured in units of mean number of band replacements per territory, this question is analogous to the question of how many times, on average, each deme changes hands between bands until full replacement. In addition, it is particularly interesting to explore how many of these band replacement events are within-species and how many are between species.

The spatial structure and the initial conditions described above lead to dynamics in which there is always a single front of interactions between the two species, to the right of which all demes are populated by Moderns, and to the left—by Neanderthals. The dynamics of the front of species interaction follow those of a random walk in 1-D with a single-absorbing state.

The number of demes in the model must reflect the carrying capacity of Europe, in terms of bands, transformed into a linear sequence of neighboring territories. This is the model's primary point of weakness: it is unclear what should be the correct transformation in order to best simulate reality in this respect. One extreme possibility would be to assume that the length of the series of demes equals the total number of bands estimated to exist in Europe. This is analogous to assuming that the spread of Moderns into all of Europe had to occur strictly along a single trajectory, as illustrated in Fig. 8a. A more realistic assumption would be that the model captures one of multiple such trajectories that occur in parallel (Fig. 8b; also a significant simplification of reality, as discussed previously). The results would be

highly sensitive to the number of these assumed independent trajectories. Since the correct choice of this number is unknown, we do not rigorously analyze the space of parameter combinations. Instead, we explore whether a range of reasonable choices for the number of demes leads to replacement durations that are of a similar order of magnitude to those found for the non-spatial model described previously, and compare the dynamics of inter-species replacement between the two models in terms of the average number of band replacements in which a band of one species replaces a band of the other.

In the reported simulations, we set the number of independent trajectories across Europe, somewhat arbitrarily, to ten (Fig. 8b), thus determining that the length of the stepping-stone chain for this model to be a tenth of Europe's carrying capacity (measured in the number of bands it supports).

**Code availability**. All simulation code and scripts for further analysis are available at https://github.com/orenkolodny.

**Data availability**. All data supporting the findings are found in the paper and in its Supplementary Notes. Unprocessed simulation outputs are available from O.K. upon request.

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

## Acknowledgements

We thank Richard Klein, Erella Hovers, Kenichi Aoki, and Victor Garcia for insightful comments and suggestions. This research is supported in part by the John Templeton Foundation and the Stanford Center for Computational, Evolutionary, and Human Genomics (CEHG).

## Author contributions

O.K. designed the study, implemented and executed the simulations, and conducted the analysis. O.K. and M.W.F. wrote the manuscript.

## Additional information

Competing interests: The authors have no competing financial interests.

