## [Peer Review File · Nature Communications]

Reviewers' comments:

Reviewer #1 (Remarks to the Author):

Manuscript "Random drift with a determined outcome...." by Kolodny and Feldman proposes a neutral population model to explain the extinction of Neanderthals. The main conclusion of the manuscript is that one can not rule out a scenario in which Moderns and Neanderthals competed neutrally on the basis of available observations. The authors further argue that the neutral model should be taken as a null model to assess the performance of alternative models.

I am not an expert in paleontology but am familiar with the mathematical model. In general, I found the subject of this manuscript fascinating and thought provoking and for these reasons it could be a great contribution for Nature Communication. On the other hand, I did not find the analysis of the robustness of the results and their comparison with the data sufficiently clear.

I have the impression that most of these problems are caused the presentation style rather than the content itself, and the whole manuscript would benefit immensely from a thorough reorganization of the manuscript, mostly in the model and results sections.

In particular, while the discussion of the model itself is quite clear to me, what I miss in the Model section are (schematic) estimates of realistic parameter ranges of the model. What degree of asymmetry in migration do the authors estimate, and why? What are reasonable number and size of bands? This is discussed in the Results section but it would be more appropriate to have this information here. Moreover, much later in the paper, it appears that the basic timestep of the paper is 25 years. Also this information should appear here and should be supported by some estimate.

Also the results suffer from a similar lack of organization. I do not understand the reason to study the average time that a band changes hand. Why not study directly the distribution of fixation times and its fluctuations? After all, it seems to me that the main quantities of interest here are the fixation time and the fixation probability. It would be also very helpful to separate the numerical results from the model from and the comparison with archaeological evidences in different subsections.

For all these reasons, I recommend a revision of this manuscript.

Reviewer #2 (Remarks to the Author):

The major claim of this paper is that modern humans could have replaced neanderthals without having any selective advantage. Although this possibility has been mentioned before, as the authors note, this is the first time that it has been developed and explored through simulation, showing that it is plausible across a wide-range of reasonable values of

key parameters. The results will be of wide interest and will certainly influence thinking in the field. The work is undoubtedly convincing and the authors themselves point that the obvious development is to give it a spatial dimension. Like them I think the result is unlikely to be qualitatively different.

The simulations are clearly presented and reproducible.

The authors show a good knowledge of the relevant archaeological and genetic literature and are right to point to the importance of the recent finding of modern human introgression in a 100,000 year old individual from the altai even if that is a long way from Europe.

I recommend publication.

Reviewer #3 (Remarks to the Author):

The manuscript submitted by O.Kolodny and M. W. Feldman proposes a scenario of migration and selectively neutral species drift to explain the replacement of Neanderthals by modern humans in Western Eurasia. The authors claim this scenario to be more parsimonious than the scenarios based on environmental causes or selective disadvantage usually entertained to explain the demise of Neanderthals. The scenario is said to be in line with archeological evidence and is tested via a mathematical model showing that under a wide range of conditions, repeated migrations and selectively neutral species drift are sufficient to explain the replacement within a length of time compatible with the archeological data.

Mathematical modeling cannot pretend to represent the reality of such a complex process, however its premises should be reasonably in line with the empirical evidence, which in my view is not the case. To start, one can question whether human populations able to develop social networks, large size coalitions and, at times, organized collective violence can simply be modeled like selectively neutral alleles randomly varying at a genetic locus. Such modeling might be operational with species of frogs migrating into a piece of jungle, but I doubt it can be used with human societies of hunter-gatherers driven by group identity and territory control. Even if one accepts the authors' methodology, the proposed model still fails to match the empirical evidence in two important aspects and its output is therefore questionable. These aspects relate to the spatial distribution of Neanderthals and modern humans, and to the notion of time of overlap between groups in a given area, which is critical in the model.

In the model, small bands of individuals can migrate from one deme to another. Deme 1 is Europe and the Levant and the Deme 2 is Africa. These groups can die and be replaced by other bands in a random way, either by individuals originating from the same deme or by migrants arriving from the other deme during the most recent time step. Under the conditions of unidirectional and constant migration, for most values of carrying capacities, the groups of migrants replace all the indigenous groups in Deme 1 after a certain period of coexistence without needing to display any selective advantage. To my understanding, the model neglects the spatial distribution of the groups within Deme 1 and migrants can freely migrate from anywhere in Africa (Deme 2) to anywhere in the Levant/Europe (Deme 1)

where the groups coexist as long as it is necessary for the complete establishment of the migrants everywhere. To say the least, this picture is far from that provided by the archeological reality. Spatial distribution and border effects might have played an important role in the replacement process.

The authors refer to various dating available in the literature to assume "10,000 to 15,000 years during which both Moderns and Neanderthals coexisted in the Levant and Europe, including a few thousand years in western Europe and including regional overlap and even recurring replacement of one species by another in particular dwelling sites". Without entering the debates surrounding many published dates and the problems arising from the identification of biological groups based on archeological evidence, the fact is that the times of overlap mentioned in the paper refer to continental coexistence between the two biological groups rather than to any proven case of local coexistence.

In Southwestern Asia, the often-entertained notion of coexistence of Neanderthals and modern humans for a long period of time is rather misleading. Modern humans are indeed documented in Israel as early as ca 120 ka ago and Neanderthals might have been present nearby until ca 50 ka BP. However, until 50 ka BP, not a single modern human is documented in western Eurasia north of the Galilee region. Meanwhile, the Southernmost found Neanderthals ever come from the Mount Carmel just south of Haifa. Therefore, between 120 ka BP and 50 ka BP, the overlap zone between the domain of early modern humans of African origin and that of western Eurasian Neanderthals is only documented in the Levant on a band of territory of 20 or 30 miles. Further discoveries might increase this distance in the future. However, at the continental scale, rather than a large area open to potential migration and admixture, we might well be dealing with a rather narrow border zone that fluctuated north to south through time.

In the middle latitudes of Eurasia, there is no evidence for the occurrence of the two biological groups in the same region for any significant period of time. North of the Levant, the first undisputed directly dated modern human comes from Ust-Ishim in Western Siberia (ca 45 ka cal BP) and is followed by an individual from Romania ca 41.4 ka cal BP. Further west, published evidence from UK and Italy ca 42-43 ka BP is rather dubious (Zilhao et al 2015; White et al. 2012). The latest directly dated Neanderthals were found in The Grotte du Renne and Saint-Césaire (center and western France) ca 41.5 ka cal BP. During this time period (40-45 ka cal BP), human groups produced various lithic assemblages representing local technical traditions. These assemblages are rather well defined and often taken as proxies for some kind of ethno-cultural entities. Some were likely made by late Neanderthals, others, particularly in central and eastern Europe, by modern migrants. Most of these assemblages are documented for several millennia within delimited territories covering hundreds of thousands of km². Although evidence for hybridization demonstrates that contact between Neanderthal and modern bands took place, hybridization events might have been restricted to the border zones and/or to periods of expansion of modern groups swamping local Neanderthals. This picture seems at odd with the large number of "changes of hand" (including changes from locals to immigrants) for each territory necessary over long periods of time to support the null hypothesis proposed by the authors (up to 1500 changes in Fig 3A). To date there is not a single site in Europe where multiple replacements

involving the two biological groups are observed. When lithic assemblages assigned to the two groups are identified in one site, the archeological evidence consistently supports a systematic replacement of the Neanderthals by modern humans with no return of the former.

Marginal remarks:

1- The authors seem to take for granted that at the time of the replacement, European Neanderthal populations were demographically weaker than modern immigrants. This is far from being demonstrated (e.g. Dogandžić & McPherron, 2013; Kuhlwilm M, et al., 2016; Richter, 2016).

2- "We find it important to address one major line of argument in this context, which suggests that Moderns had a cognitive and cultural advantage, potentially in the form of symbolic thought or language, over Neanderthals (9, 44–49, 72, 85). To date, genetic and cranio- morphological comparisons between the species have not produced any unequivocal evidence that would support this argument". What about Prüfer et al. (2014) or McCoy et al (2017)?

3- What, in the view of the authors, triggered the migration of modern humans out of "Africa" (including the south of the Levant and the Arabian Peninsula)? Is it only demography? If this is true, isn't it reasonable to assume that this large African population experienced a higher rate of cultural innovation and was submitted to stronger selection pressure toward more cognitive complexity?

References:

Dogandžić T, McPherron SP. 2013, Demography and the demise of Neandertals: a comment on 'Tenfold population increase in Western Europe at the Neandertal-to-modern human transition'. *J Hum Evol.* 64(4):311-3.

Kuhlwilm M, et al. (2016) Ancient gene flow from early modern humans into Eastern Neanderthals. *Nature* 530(7591):429–433.

McCoy R. et al. (2017) Impacts of Neanderthal-Introgressed Sequences on the Landscape of Human Gene Expression. *Cell* 168 (5): 916–927

Prüfer, K et al. (2014). The complete genome sequence of a Neanderthal from the Altai Mountains. *Nature*, 505(7481), 43-49.

Richter J. (2016), Leave at the height of the party: A critical review of the Middle Paleolithic in Western Central Europe from its beginnings to its rapid decline *Quaternary International* 411: 107–128

White M, Pettitt P. (2012) Ancient Digs and Modern Myths: The Age and Context of the

Kent's Cavern 4 Maxilla and the Earliest Homo sapiens Specimens in Europe. *Eur J Archaeol.* 15: 392–420.

Zilhão J, Banks WE, d'Errico F, Gioia P (2015) Analysis of Site Formation and Assemblage Integrity Does Not Support Attribution of the Uluzzian to Modern Humans at Grotta del Cavallo. *PLoS ONE* 10(7): e0131181.

May 2017

Detailed response to reviewers' comments

We sincerely appreciate the close reading and thorough assessment of the manuscript by all three reviewers. We have found the reviewers' comments constructive, and believe our revised version addresses their concerns comprehensively. Below we respond to these comments in detail and refer to respective changes made in the manuscript, including the addition of a supplementary section in which we describe a new model of species' replacement and its results. This model includes a spatial component, addressing the major concerns raised by reviewer #3.

Reviewers' comments:

Reviewer #1 (Remarks to the Author):

Manuscript "Random drift with a determined outcome...." by Kolodny and Feldman proposes a neutral population model to explain the extinction of Neanderthals. The main conclusion of the manuscript is that one can not rule out a scenario in which Moderns and Neanderthals competed neutrally on the basis of available observations. The authors further argue that the neutral model should be taken as a null model to assess the performance of alternative models.

I am not an expert in paleontology but am familiar with the mathematical model. In general, I found the subject of this manuscript fascinating and thought provoking and for these reasons it could be a great contribution for Nature Communication.

Thank you!

On the other hand, I did not find the analysis of the robustness of the results and their comparison with the data sufficiently clear. I have the impression that most of these problems are caused the presentation style rather than the content itself, and the whole manuscript would benefit immensely from a thorough reorganization of the manuscript, mostly in the model and results sections.

We have made changes following the suggestions below in order to address this concern, and feel that the manuscript is greatly improved. See details below. Thank you!

In particular, while the discussion of the model itself is quite clear to me, what I miss in the Model section are (schematic) estimates of realistic parameter ranges of the model. What degree of asymmetry in migration do the authors estimate, and why? What are reasonable number and size of bands? This is discussed in the Results section but it would be more appropriate to have this information here. Moreover, much later in the paper, it appears that the basic timestep of the paper is 25 years. Also this information should appear here and should be supported by some estimate.

This and the following comment have been addressed by reorganizing the manuscript: the results section which included the explanation of the hypothesis rejection test has been moved to the Model section, with some added clarifications.

In particular, we have highlighted which parameters' values need to be assumed in order to derive the model's numerical results and which need to be assumed later, in order to compare the results to the archaeological record. Because these parameter assumptions are highly debated, it was important for us to avoid "baking" these assumptions into the model itself, and the model is structured such that few assumptions need to be made in the implementation of the model and the derivation of numerical results from it, relegating the arguable assumptions to the post-simulation application of the hypothesis test. This allows the reader to consider our model in light of the specific parameter estimates from the literature that he/she finds most reliable. Your comment has helped us realize that this needed further clarification, which we have now carried out.

The Model section now includes a presentation of the range of parameters estimated from the literature for band numbers and sizes; these are also called upon in a later result section to demonstrate the hypothesis test based on the provided figure.

The order of figures 2 and 3 has been switched, to reflect their relative importance to the argument we make and to improve readability.

We have added references in the model section where we present our justification for focusing on unidirectional migration, and now note explicitly that bi-directional migration (both symmetric and a-symmetric) is provided for the sake of completeness, although only few researchers in the field perceive this possibility as realistic. We are agnostic regarding which scenario is more realistic, but it seems that the absence of migration of Neanderthals into Africa is one of the few points of near-agreement in the literature regarding the Neanderthal-Modern dynamics.

The figure of 25 years refers to the length of a generation, which is the unit (not the parameter value) that we use; in the context in which it is now presented, we believe this would be clearer.

Also the results suffer from a similar lack of organization. I do not understand the reason to study the average time that a band changes hand. Why not study directly the distribution of fixation times and its fluctuations? After all, it seems to me that the main quantities of interest here are the fixation time and the fixation probability.

The explanation regarding the interpretation of figure 3 (now figure 2B) was unclear, and has been changed significantly, addressing this concern.

We study fixation times directly as the reviewer proposes (Figure 1A), as well as species co-habitation time until replacement (Figure 2B), which is similar but more relevant, in a sense: it measures the analogue of the fixation time, but starts the measurement not from the beginning of the simulation but from the beginning of the period in which the two species' presence is likely to be detected in Europe. The units in which this time period is presented in Figure 2B are the number of changes-of-hand-per-band-territory,

because this unit is more independent of literature-derived assumptions (as discussed above) than a translation of this numerical result to years. It is also easier to interpret in realistic terms than using the term “model time step”, thus supporting an easier interpretation of figure 2B which presents the results of the hypothesis test for each point in the space of parameter value combinations.

It would be also very helpful to separate the numerical results from the model from and the comparison with archaeological evidences in different subsections.

We have left these results in a common section, but split their discussion into separate paragraphs and changed the wording to highlight clearly which result is a numerical result and which is a comparison to the literature.

This section has also been shortened significantly, so it is much more readable and orderly.

Reviewer #2 (Remarks to the Author):

The major claim of this paper is that modern humans could have replaced neanderthals without having any selective advantage. Although this possibility has been mentioned before, as the authors note, this is the first time that it has been developed and explored through simulation, showing that it is plausible across a wide-range of reasonable values of key parameters. The results will be of wide interest and will certainly influence thinking in the field. The work is undoubtedly convincing and the authors themselves point that the obvious development is to give it a spatial dimension. Like them I think the result is unlikely to be qualitatively different.

The simulations are clearly presented and reproducible.

The authors show a good knowledge of the relevant archaeological and genetic literature and are right to point to the importance of the recent finding of modern human introgression in a 100,000 year old individual from the altai even if that is a long way from Europe.

I recommend publication.

Thank you!

Reviewer #3 (Remarks to the Author):

The manuscript submitted by O.Kolodny and M. W. Feldman proposes a scenario of migration and selectively neutral species drift to explain the replacement of Neanderthals by modern humans in Western Eurasia. The authors claim this scenario to be more parsimonious than the scenarios based on environmental causes or selective disadvantage usually entertained to explain the demise of Neanderthals. The scenario is said to be in line with archeological evidence and is tested via a mathematical model showing that under a wide range of conditions, repeated migrations and selectively

neutral species drift are sufficient to explain the replacement within a length of time compatible with the archeological data.

Mathematical modeling cannot pretend to represent the reality of such a complex process, however its premises should be reasonably in line with the empirical evidence, which in my view is not the case.

We sincerely appreciate the reviewer's careful reading of our paper and the detailed comments. Addressing them has significantly improved our manuscript: they have highlighted a number of aspects that required explicit discussion or clarifications, which we have now added in the main text and in two additional supplementary sections. We agree with the reviewer on many points, and his comments helped us find and remedy problems in the text. To address the reviewer's primary concern – that the lack of a spatial component in the model may lead to significant deviation from what should be our realistic expectations regarding the species' replacement dynamics – we have implemented a new model that includes a spatial component. A full treatment of the spatial complexity of the dynamics, as is discussed in the main text and supplementary section C, requires complex modeling and a large number of strong assumptions, making the model less general and more sensitive to specific details about which there is no agreement among researchers of the field. Such treatment of the topic is an important future avenue of research, well beyond the scope of our current study. In line with the goal of the current study, we have constructed the simplest model that would qualitatively capture the effect of a spatial component in the inter-species dynamics of interest, such that it would be tractable, intuitive, and would require a minimal number of assumptions. This model's results differ significantly from those of our primary model in the number of inter-species "change of hands" of each territory, reducing it significantly. This addresses the reviewer's second major concern. These are discussed in detail below and in the added supplementary section, section D.

To start, one can question whether human populations able to develop social networks, large size coalitions and, at times, organized collective violence can simply be modeled like selectively neutral alleles randomly varying at a genetic locus. Such modeling might be operational with species of frogs migrating into a piece of jungle, but I doubt it can be used with human societies of hunter-gatherers driven by group identity and territory control.

- Competition (including inter-group violence and inter-group competition) between bands, based on band identity and control of a resource (territory or other) is completely in line with our model's assumptions; the Moran model we use doesn't assume that these interactions do not occur; it merely assumes that the outcome of each competition (i.e. which of the competing bands will die out) is independent of the species identity of that band. This is now made explicit in the text, lines 125-135. We have also added a discussion of the topic and related issues in supplementary section C, subsection 4 (see also below).

- The parameter value used for the mean rate of band replacement should be one that takes into account the mean effect of such inter-band competition. The rates that are cited in the Results section 1.2, based on the rate inferences in ((Soltis, Boyd, & Richerson, 1995)), are derived from anthropological accounts of small traditional groups with individualized group identities that conduct warfare and have replaced one another in recent times. As discussed, these are far from representative of the Neanderthals and Moderns 40kya, but are the best available, as far as we know.
- Ecological modeling with regard to human behavior and human dynamics has been done extensively and successfully in the past (e.g. (Banks et al., 2006, 2008; Belovsky, 1988; Gilpin, Feldman, & Aoki, 2016; Winterhalder & Smith, 2000)). We suggest it is a productive approach, as long as it is applied carefully. In particular, we suggest it is a useful approach as a null model of expected population dynamics.
- It has been proposed that species' differences in the ability/tendency to create large social networks and coalitions may have played a role in the Neanderthals' replacement (e.g. (Gat, 1999)), but there is no unambiguous evidence in support of this suggestion. Moreover, we have no means of knowing what the coalition and violence dynamics were, and it is not clear how such behavior would bias the process (whether they would shorten or elongate it, for example). Although such dynamics may have transpired, and the exploration of such possible dynamics is interesting, it is a scenario that is based on more assumptions than our model, and we believe it is reasonable to avoid these when constructing a parsimonious null model. We now discuss this topic in Supplementary section C.4.
- We think that most people's intuition would be to assume that coalitions of this sort would tend to be of bands of one species against bands, or coalitions of bands, of the other species. Historical accounts of modern ethnic groups that expand to new regions generally do not support this intuition: when European colonialists spread through the Americas, for example, they fought amongst them repeatedly, created coalitions with native groups against other native groups, against other Europeans, and against opposing mixed coalitions (see, (Matthew & Oudijk, 2014; Restall, 2004); obviously, these are within the same species, but demonstrate coalitions that do not necessarily follow lines of ethnic relatedness). Similar dynamics were common in the colonial era in Africa (e.g. (Vandervort, 1998)). This suggests that even if coalitions are considered, there is no a-priori reason to believe that they were uniquely within one species and against the other; thus, there are no grounds to assume, in a null model, that coalitions would produce a skew in the expected dynamics towards shorter or longer replacement times compared to those predicted by our model.

Even if one accepts the authors' methodology, the proposed model still fails to match the empirical evidence in two important aspects and its output is therefore

questionable. These aspects relate to the spatial distribution of Neanderthals and modern humans, and to the notion of time of overlap between groups in a given area, which is critical in the model.

We agree that these are important factors, but suggest that it is valuable to explore a null model that does not include spatial distribution. Spatial structure is discussed in the manuscript and in supplementary section C, and is mentioned explicitly as an important avenue for future exploration. We do not think such explicit incorporation would yield qualitatively different results from those of our current model, and have now added a simple model of species replacement that includes a spatial component, largely confirming this prediction (see supplementary section D).

Incorporation of spatial distribution is, as the reviewer implies, tied into time of overlap as well, and calls for a different type of modeling, such as that of diffusion waves or of a shifting front of interspecies interaction as in the model we now added. This is part of the reason that we address an overall period of overlap in the entirety of Europe or Eurasia in our current (main text) model. Using regional estimates of time of overlap is incompatible with a model that does not consider regional dynamics explicitly. This is addressed in more detail below, and we have now incorporated a more explicit discussion of this point in supplementary section A.2.

In addition, to address this major concern, we have now added a second model of species replacement that includes a spatial component. It is highly simplified, as a full treatment of spatial dynamics is well beyond the scope of the current study, but it serves to demonstrate that our qualitative findings hold in a spatially explicit setting, to address some specific concerns raised in the comments below, and to highlight some of the main differences between a spatially explicit model and the non-spatial null model. This is discussed in detail below and in the new supplementary section D.

In the model, small bands of individuals can migrate from one deme to another. Deme 1 is Europe and the Levant and the Deme 2 is Africa. These groups can die and be replaced by other bands in a random way, either by individuals originating from the same deme or by migrants arriving from the other deme during the most recent time step. Under the conditions of unidirectional and constant migration, for most values of carrying capacities, the groups of migrants replace all the indigenous groups in Deme 1 after a certain period of coexistence without needing to display any selective advantage. To my understanding, the model neglects the spatial distribution of the groups within Deme 1 and migrants can freely migrate from anywhere in Africa (Deme 2) to anywhere in the Levant/Europe (Deme 1) where the groups coexist as long as it is necessary for the complete establishment of the migrants everywhere.

This is correct. Notably, either species can go extinct in deme 1; in fact, Moderns typically establish in Europe but then go extinct multiple times in each simulation, before the establishment event of the lineage that eventually increases in frequency and reaches fixation.

To say the least, this picture is far from that provided by the archeological reality.

Agreed. This is an obvious simplification: any population model that assumes within-deme panmixia is necessarily a simplification of reality. This simplification is common to the vast majority of evolutionary, genetic, and population dynamics' models.

Spatial distribution and border effects might have played an important role in the replacement process.

We agree, but it is important to explore the dynamics of a simple null model that does not consider these realistic complications.

The model that we have now added to address these concerns demonstrates some prominent differences and similarities between spatial and non-spatial models of the replacement dynamics (see below).

The authors refer to various dating available in the literature to assume "10,000 to 15,000 years during which both Moderns and Neanderthals coexisted in the Levant and Europe, including a few thousand years in western Europe and including regional overlap and even recurring replacement of one species by another in particular dwelling sites". Without entering the debates surrounding many published dates and the problems arising from the identification of biological groups based on archeological evidence, the fact is that the times of overlap mentioned in the paper refer to continental coexistence between the two biological groups rather than to any proven case of local coexistence.

Agreed; this was our intent, given the model's assumption of within-deme panmixia. This is now noted in supplementary section A.2.

In Southwestern Asia, the often-entertained notion of coexistence of Neanderthals and modern humans for a long period of time is rather misleading. Modern humans are indeed documented in Israel as early as ca 120 ka ago and Neanderthals might have been present nearby until ca 50 ka BP. However, until 50 ka BP, not a single modern human is documented in western Eurasia north of the Galilee region. Meanwhile, the Southernmost found Neanderthals ever come from the Mount Carmel just south of Haifa. Therefore, between 120 ka BP and 50 ka BP, the overlap zone between the domain of early modern humans of African origin and that of western Eurasian Neanderthals is only documented in the Levant on a band of territory of 20 or 30 miles. Further discoveries might increase this distance in the future. However, at the continental scale, rather than a large area open to potential migration and admixture, we might well be dealing with a rather narrow border zone that fluctuated north to south through time.

True; although given the sparseness of the record in this regions from these periods, it would not be surprising if this border zone will increase significantly in the future. To some, the finding of Modern introgression in Altai Neanderthals is suggestive of a

greater geographical overlap that includes northern regions as well (even though this is not necessary). In any case, these are some of the reasons that, in the estimates we used, we did not consider the potentially extremely long period of overlap in the Levant. This is discussed and pointed out as an important aspect for future exploration in supplementary section C, subsections 2 and 3.

Naturally, if one does consider this long period of potential overlap, it becomes clear that the duration of replacement was well within the range predicted by neutral dynamics.

In the middle latitudes of Eurasia, there is no evidence for the occurrence of the two biological groups in the same region for any significant period of time. North of the Levant, the first undisputed directly dated modern human comes from Ust-Ishim in Western Siberia (ca 45 ka cal BP) and is followed by an individual from Romania ca 41.4 ka cal BP. Further west, published evidence from UK and Italy ca 42-43 ka BP is rather dubious (Zilhao et al 2015; White et al. 2012). The latest directly dated Neanderthals were found in The Grotte du Renne and Saint-Césaire (center and western France) ca 41.5 ka cal BP.

We agree. This topic is discussed in some detail at the last end of Supplementary section A. These findings have been taken into account:

1. Since Moderns had to reach Siberia somehow, one should expect that they were found North of the Levant at least some time before 45ka. Given Higham et al.'s (2014) slightly later estimate of the timing of the latest Neanderthals, at circa 40ka, we suggest that a reasonable minimalistic period of both species' existence in Europe is 5000 years. Figure S3 provides an analysis analogous to that found in the main text, but assuming a duration of only 5000 years for species replacement. The results of this analysis are qualitatively similar to those reported in the main text: for a broad range of reasonable parameter combinations, the null model is not rejected in favor of a selective scenario.
2. The slightly longer 10ky-15ky estimate which is used in the main text, as explained in supplementary section A, assumes that it is correct to consider the Moderns' remains in the Levant which are found fairly close to the period of replacement (54kya, as opposed to those found there 120kya) as a part of the period of species overlap.

During this time period (40-45 ka cal BP), human groups produced various lithic assemblages representing local technical traditions. These assemblages are rather well defined and often taken as proxies for some kind of ethno-cultural entities. Some were likely made by late Neanderthals, others, particularly in central and eastern Europe, by modern migrants. Most of these assemblages are documented for several millennia within delimited territories covering hundreds of thousands of km². Although evidence for hybridization demonstrates that contact between Neanderthal and modern bands took place, hybridization events might have been restricted to the border zones and/or to periods of expansion of modern groups swamping local Neanderthals.

These “transitional” techno-complexes are among the most interesting observations in the Modern-Neanderthal dynamics. We have very recently written a paper that suggests a possible qualitative explanation of the underlying cultural-evolutionary dynamics that led to the creation of these assemblages (Creanza, Kolodny, & Feldman, 2017a).

This picture seems at odd with the large number of “changes of hand” (including changes from locals to immigrants) for each territory necessary over long periods of time to support the null hypothesis proposed by the authors (up to 1500 changes in Fig 3A). To date there is not a single site in Europe where multiple replacements involving the two biological groups are observed. When lithic assemblages assigned to the two groups are identified in one site, the archeological evidence consistently supports a systematic replacement of the Neanderthals by modern humans with no return of the former.

We see your point, and to address this concern we have done a number of things:

1. We added a figure with the mean number of inter-species band replacements in Supplementary section A together with a discussion of the topic.
2. We have added the spatially-explicit model: the number of inter-species band replacement events is the major aspect in which the results of the new model (supplementary section D) deviate from those of the model in the main text. It shows that when the distribution of bands along the landscape results from a gradually shifting front of inter-species interaction, the vast majority of replacements are within-species, because each band is surrounded most of the time by bands of its own species and is replaced by one of them when it dies out. In the spatially explicit model the mean number of inter-species band replacements per site is on the order of 4 to 13, depending on the population size, and in many simulation instances the number of inter-species band replacements per site is even lower. The topic is elaborated further in the Discussion of supplementary section D.
3. We have added a supplementary section (E) that includes a rough “back-of-the-envelope” calculation, which suggests that given the sparseness of the archaeological record, it is to be expected that direct evidence for recurring replacement or for contemporaneity of the two species at any single site would be scarce or completely absent. This complex issue deserves more detailed analysis, which is beyond the scope of the current study. The informal calculation in the added section serves merely to demonstrate the type of analysis we think might be useful in this context.

Marginal remarks:

1- The authors seem to take for granted that at the time of the replacement, European Neanderthal populations were demographically weaker than modern immigrants. This is far from being demonstrated (e.g. Dogandžić & McPherron, 2013; Kuhlwilm M, et al., 2016; Richter, 2016).

This comment suggests that the assumptions of our model (or, more precisely, their interpretation and the interpretation of the model's dynamics) were not sufficiently clear, and we have now made changes in the Discussion accordingly (lines 501-508). The point raised by the reviewer and the perspective suggested by the listed references are completely in line with, and in fact support, the dynamics that we suggest may have taken place: our model does not suggest a swamping scenario; instead, it suggests that a slow, discrete, "trickle" of small groups of Moderns into Eurasia would have been sufficient to determine the eventual replacement of Neanderthals, as long as this trickle is assumed to have been unidirectional, only out of Africa (section 1 of the results, *unidirectional migration*, which presents the main result of the paper; this assumption is discussed below). In the model explored in section 1 of the results, one or few of the small bands that arrive in this slow trickle happen to establish, and eventually one or a few of them stochastically and gradually increase in frequency through selectively neutral drift. This does not necessitate an assumption of any demographic superiority of Moderns. Moreover, it demonstrates that such superiority is not necessary to explain the replacement. In the model explored in section 1 of our results (*unidirectional migration*), the initial population sizes of the two species plays no role in determining the dynamics.

We have now added a reference to Dogandžić & McPherron, 2013, and we previously referred to Kuhlwilm M, et al., 2016 in support of a suggestion similar in spirit to the one implied by the reviewer: that the finding of early introgression of Neanderthals into Moderns suggests an early phase in which the two species interacted while each of them constituted significant portions of the population, at least locally.

2- "We find it important to address one major line of argument in this context, which suggests that Moderns had a cognitive and cultural advantage, potentially in the form of symbolic thought or language, over Neanderthals (9, 44–49, 72, 85). To date, genetic and cranio- morphological comparisons between the species have not produced any unequivocal evidence that would support this argument". What about Prüfer et al. (2014) or McCoy et al (2017)?

We are aware of these papers (and had cited one of them), and although we find them very important to our understanding of the dynamics, we don't interpret any of their findings as evidence of a general cognitive advantage of Moderns over Neanderthals. We may have misunderstood this comment.

- Perhaps what the reviewer meant was that the down-regulation of Neanderthal genes in present-day Moderns' brains might be interpreted as reflecting a cognitive disadvantage prescribed by these genes. We suggest that such an interpretation is unwarranted: these genes' down-regulation suggests that in their current context, within the genomic landscape of a primarily-Modern genome, they are disadvantageous, but this does not imply that they would have

been detrimental in the context of a primarily-Neanderthal genome, and provides no information about the general cognitive capacities of Neanderthals compared to Moderns).

- A similar argument can be made regarding possibly higher inbreeding among (some) Neanderthals: a high inbreeding coefficient, or even just a small population size, could have led to high deleterious mutational load, but does not imply necessarily a reduced cognitive capacity.

3- What, in the view of the authors, triggered the migration of modern humans out of “Africa” (including the south of the Levant and the Arabian Peninsula)? Is it only demography? If this is true, isn’t it reasonable to assume that this large African population experienced a higher rate of cultural innovation and was submitted to stronger selection pressure toward more cognitive complexity?

These are very interesting questions, which are related to other studies of ours, including two forthcoming papers on the effects of culture on human ecology and evolution (Creanza, Kolodny, & Feldman, 2017b) and on the effect of culture on the evolution of cognition (Lotem, Halpern, Edelman, & Kolodny, 2017), a recent paper that explores the possibility of culturally-driven replacement (non-neutral, as opposed to the null model presented in the current study; (Gilpin et al., 2016)), and two very recent studies that use a computational cultural evolution framework to explore, among other factors, the relation between population size, population interactions, cultural innovation, and feedback loops among these (Creanza et al., 2017a; Kolodny, Creanza, & Feldman, 2016).

The relation between population size and cultural complexity is hotly debated (see discussion and references found in (Creanza et al., 2017a; Kolodny et al., 2016)), and the mode and rate in which selective pressures, including density-related competition, change cognitive capacities is largely unknown (Lotem et al., 2017). We suggest that although these factors *may* have played a meaningful role, it is extremely important to start by considering a simpler scenario that explores what we expect would have transpired if none of these less parsimonious dynamics/factors were present and without assuming a cognitive advantage to Moderns as a result of demographic factors or others.

The question of what led to the migration dynamics out of Africa is paramount to our full understanding of the Middle to Upper Paleolithic transition. Among the possible drivers of outgoing migration could be:

- Stochastic fluctuations in population density, which led to periods of increased competition and higher motivation to migrate.
- Local or small-scale climatic change that created favorable conditions, leading to increased population densities that in turn spurred increased migration.
- Local deterioration in environmental conditions, spurring migration as a means of escaping the poor conditions (interestingly, what matters - in this scenario and possibly also in the previous one - may be not the average state of the environmental conditions, but the variance in these conditions between years or

between successive generations, where high variance may be more likely to lead to repeated, transient, small waves of outgoing migration).

- Cultural changes in practices related to resource procurement and subsistence strategy (more or less sedentary, for example).

In this study we intentionally remain agnostic regarding the drivers of this migration, and accept the archaeological observation at face value: it seems that migration occurred only out of Africa and not into it, in at least a number of migration events. We suggest that it is reasonable to model this as a slow rate of discrete events of unidirectional migration.

References:

Dogandžić T, McPherron SP. 2013, Demography and the demise of Neandertals: a comment on 'Tenfold population increase in Western Europe at the Neandertal-to-modern human transition'. *J Hum Evol.* 64(4):311-3.

Kuhlwilm M, et al. (2016) Ancient gene flow from early modern humans into Eastern Neanderthals. *Nature* 530(7591):429–433.

McCoy R. et al. (2017) Impacts of Neanderthal-Introgressed Sequences on the Landscape of Human Gene Expression. *Cell* 168 (5): 916–927

Prüfer, K et al. (2014). The complete genome sequence of a Neanderthal from the Altai Mountains. *Nature*, 505(7481), 43-49.

Richter J. (2016), Leave at the height of the party: A critical review of the Middle Paleolithic in Western Central Europe from its beginnings to its rapid decline *Quaternary International* 411: 107–128

White M, Pettitt P. (2012) Ancient Digs and Modern Myths: The Age and Context of the Kent's Cavern 4 Maxilla and the Earliest Homo sapiens Specimens in Europe. *Eur J Archaeol.* 15: 392–420.

Zilhão J, Banks WE, d'Errico F, Gioia P (2015) Analysis of Site Formation and Assemblage Integrity Does Not Support Attribution of the Uluzzian to Modern Humans at Grotta del Cavallo. *PLoS ONE* 10(7): e0131181.

Thank you! We have added some of these citations to our manuscript.

References

- Banks, W. E., d'Errico, F., Dibble, H. L., Krishtalka, L., West, D., Olszewski, D. I., ... Montet-White, A. (2006). Eco-cultural niche modeling: new tools for reconstructing the geography and ecology of past human populations. *PaleoAnthropology*, 4, 68–83.
- Banks, W. E., d'Errico, F., Peterson, A. T., Vanhaeren, M., Kageyama, M., Sepulchre, P., ... Lunt, D. (2008). Human ecological niches and ranges during the LGM in Europe derived from an application of eco-cultural niche modeling. *Journal of Archaeological Science*, 35(2), 481–491.
- Belovsky, G. E. (1988). An optimal foraging-based model of hunter-gatherer population dynamics. *Journal of Anthropological Archaeology*, 7(4), 329–372.
- Creanza, N., Kolodny, O., & Feldman, M. (2017a). Greater than the sum of its parts? Modeling population contact and interaction of cultural repertoires. *Journal of the Royal Society, Interface / the Royal Society*, (In Press).
- Creanza, N., Kolodny, O., & Feldman, M. W. (2017b). Cultural evolutionary theory: how culture evolves and why it matters. *Proc. Natl. Acad. Sci.*, (Accepted, under minor revision).
- Gat, A. (1999). Social organization, group conflict and the demise of Neanderthals. *Mankind Quarterly*, 39(4), 437.
- Gilpin, W., Feldman, M. W., & Aoki, K. (2016). An ecocultural model predicts Neanderthal extinction through competition with modern humans. *Proceedings of the National Academy of Sciences*, 113(8), 2134–2139.
<http://doi.org/10.1073/pnas.1524861113>
- Kolodny, O., Creanza, N., & Feldman, M. W. (2016). Game-changing innovations: how culture can change the parameters of its own evolution and induce abrupt cultural shifts. *Submitted*.
- Lotem, A., Halpern, J., Edelman, S., & Kolodny, O. (2017). Culture and the evolution of cognition: a process-level approach. *Proc. Natl Acad. Sci. USA*, *Accepted*.
- Matthew, L. E., & Oudijk, M. R. (2014). *Indian conquistadors: Indigenous allies in the conquest of Mesoamerica*. University of Oklahoma Press.
- Restall, M. (2004). *Seven myths of the Spanish conquest*. Oxford University Press.
- Soltis, J., Boyd, R., & Richerson, P. J. (1995). Can group-functional behaviors evolve by cultural group selection?: An empirical test. *Current Anthropology*, 36(3), 473–494.
- Vandervort, B. (1998). *Wars of imperial conquest in Africa, 1830-1914*. Indiana University Press.
- Winterhalder, B., & Smith, E. A. (2000). Analyzing adaptive strategies: Human behavioral ecology at twenty-five. *Evolutionary Anthropology Issues News and Reviews*, 9(2), 51–72.

REVIEWERS' COMMENTS:

Reviewer #1 (Remarks to the Author):

In the revised version, the authors properly addressed all my concerns and, to the best of my understanding, those of the other referees. In particular, I feel that the clarity of the manuscript has greatly improved with this revision.

This manuscript presents an interesting model raising a thought-provoking hypothesis that I believe can have an important impact in this field. Therefore, I recommend publication.

Reviewer #3 (Remarks to the Author):

The authors have thoroughly addressed the remarks by the reviewers of the paper. Although one can question whether this modelling provides a good representation of the reality of the Neandertal/modern human replacement in Western Eurasia, it has the merit of testing the hypothesis of neutral species drift to explain it. In a nutshell, constant migration (for whatever reason) of modern humans out of Africa into the Neandertal domain would result in the final replacement of indigenous populations, even in the absence of selective advantage on the side of the immigrants. In this scenario, the number of band replacements at a given place within the Eurasian deme is critical to validate the model. In this respect, I appreciate the more cautious approach of the authors, in particular in the discussion section.

Importantly, the authors added a model into the supplementary information that takes into account population spatial structures. Interestingly, the aspect in which the results of the spatially explicit model differs the most from those of the non-spatial model, is the number of inter-species band replacements per band of territory.

I again warn the authors that "recurring replacement of one species by another in particular dwelling sites" is not documented in a convincing way in a single European site. Among the references quoted in the significance statement (refs. 18-28), the only one referring to such a phenomenon, Gravina et al 2005, has been heavily criticised.

Although I still stand on the side of those who believe that selective or demographic advantage played a role in the expansion of modern humans over the planet, not only replacing Neandertals, but any other forms of archaic humans, I find the paper very interesting, as it allows future tests in light of archaeological and paleogenetic evidence. It is quite interesting that in the proposed model, Moderns' frequency can decrease after reaching intermediate values, which might well have been the case in western Eurasia, as far as current archaeological data allow us to track these early phases of modern immigration.

Reviewer #1 (Remarks to the Author):

In the revised version, the authors properly addressed all my concerns and, to the best of my understanding, those of the other referees. In particular, I feel that the clarity of the manuscript has greatly improved with this revision.

This manuscript presents an interesting model raising a thought-provoking hypothesis that I believe can have an important impact in this field. Therefore, I recommend publication.

Thank you very much!

Reviewer #3 (Remarks to the Author):

The authors have thoroughly addressed the remarks by the reviewers of the paper. Although one can question whether this modelling provides a good representation of the reality of the Neandertal/modern human replacement in Western Eurasia, it has the merit of testing the hypothesis of neutral species drift to explain it. In a nutshell, constant migration (for whatever reason) of modern humans out of Africa into the Neandertal domain would result in the final replacement of indigenous populations, even in the absence of selective advantage on the side of the immigrants. In this scenario, the number of band replacements at a given place within the Eurasian deme is critical to validate the model. In this respect, I appreciate the more cautious approach of the authors, in particular in the discussion section.

Importantly, the authors added a model into the supplementary information that takes into account population spatial structures. Interestingly, the aspect in which the results of the spatially explicit model differs the most from those of the non-spatial model, is the number of inter-species band replacements per band of territory.

I again warn the authors that "recurring replacement of one species by another in particular dwelling sites" is not documented in a convincing way in a single European site. Among the references quoted in the significance statement (refs. 18-28), the only one referring to such a phenomenon, Gravina et al 2005, has been heavily criticised.

Thank you. We have attenuated the sentence that refers to this topic.

Although I still stand on the side of those who believe that selective or demographic advantage played a role in the expansion of modern humans over the planet, not only replacing Neandertals, but any other forms of archaic humans, I find the paper very interesting, as it allows future tests in light of archaeological and paleogenetic evidence. It is quite interesting that in the proposed model, Moderns' frequency can decrease after reaching intermediate values, which might well have been the case in western Eurasia, as far as current archaeological data allow us to track these early phases of modern immigration.

Thank you very much for the thorough consideration and comments!